# Inner nuclear protein Matrin-3 coordinates cell differentiation by stabilizing chromatin architecture

Hye Ji Cha[1], Özgün Uyan[2,11], Yan Kai [3,11], Tianxin Liu[1], Qian Zhu [1], Zuzana Tothova[4,5,6], Giovanni A. Botten [7], Jian Xu [7], Guo-Cheng Yuan[3], Job Dekker [8,9] & Stuart H. Orkin [1,10✉]

Precise control of gene expression during differentiation relies on the interplay of chromatin and nuclear structure. Despite an established contribution of nuclear membrane proteins to developmental gene regulation, little is known regarding the role of inner nuclear proteins. Here we demonstrate that loss of the nuclear scaffolding protein Matrin-3 (Matr3) in erythroid cells leads to morphological and gene expression changes characteristic of accelerated maturation, as well as broad alterations in chromatin organization similar to those accompanying differentiation. Matr3 protein interacts with CTCF and the cohesin complex, and its loss perturbs their occupancy at a subset of sites. Destabilization of CTCF and cohesin binding correlates with altered transcription and accelerated differentiation. This association is conserved in embryonic stem cells. Our findings indicate Matr3 negatively affects cell fate transitions and demonstrate that a critical inner nuclear protein impacts occupancy of architectural factors, culminating in broad effects on chromatin organization and cell differentiation.

[1] Division of Hematology/Oncology, Boston Children's Hospital and Department of Pediatric Oncology, Dana-Farber Cancer Institute (DFCI), Harvard Stem Cell Institute, Harvard Medical School, Boston, MA, USA. [2] Department of Neurology, University of Massachusetts Medical School, Worcester, MA, USA. [3] Department of Pediatric Oncology, Dana-Farber Cancer Institute and Harvard Medical School, Boston, MA, USA. [4] Department of Medical Oncology, Dana-Farber Cancer Institute, Boston, MA, USA. [5] Division of Hematology, Brigham and Women's Hospital, Boston, MA, USA. [6] Broad Institute of MIT and Harvard, Cambridge, MA, USA. [7] Children's Medical Center Research Institute, Department of Pediatrics, University of Texas Southwestern Medical Center, Dallas, TX, USA. [8] Program in Systems Biology, Department of Biochemistry and Molecular Pharmacology, University of Massachusetts Medical School, Worcester, MA, USA. [9] Howard Hughes Medical Institute, University of Massachusetts Medical School, Worcester, MA, USA. [10] Howard Hughes Medical Institute, Boston, MA, USA. [11] These authors contributed equally: Özgün Uyan, Yan Kai. ✉email: stuart_orkin@dfci.harvard.edu

The nucleus is spatially organized by chromosome and interchromatin functional components. Recent advances in genome-wide analysis of chromosome conformation have provided molecular information regarding chromosome folding, and partitioned the genome into two compartments. The A and B compartments correspond to the structures and characteristics of known euchromatin and heterochromatin, respectively[1,2]. Recent biophysical studies suggest that distinct chromatin regions may be pulled together or move away from each other by phase-separated nuclear condensates. For example, droplets of heterochromatin protein 1 (HP1) facilitate sequestration of compacted chromatin, which may result in steric exclusion of regulatory proteins, such as RNA polymerase, from the underlying DNA[3,4]. In active regions of the genome, transcription factors and co-activators form condensates that compartmentalize the transcription machinery and drive gene activation[5–7]. Global reorganization of chromatin interactions and compartmentalization occurring during differentiation[8] requires proper chromosome positioning, but the involvement of nuclear components in this process is unknown.

Architectural proteins play a critical role in chromatin organization and function. Two well-characterized proteins, CCCTC-binding factor (CTCF) and cohesin, organize topological chromatin domains and mediate chromatin interactions of individual genomic loci[9,10]. At the nuclear periphery, a meshwork of lamina proteins provides anchoring sites for genomic loci, and attachment is often accompanied by gene inactivation[11]. Conversely, detachment from the nuclear periphery frequently correlates with gene activation, reflecting counterforces generated by intra-nuclear substructures. Nuclear speckles, for example, act as functional centers that organize active gene loci to form euchromatic districts[12,13]. In addition, abundant nucleoplasmic proteins serve as structural scaffolds spanning the nucleus, and specific inner nuclear proteins have been implicated in maintenance of eu- and heterochromatin architecture[14,15].

Coordinated regulation of spatial and temporal chromatin repositioning is important for proper gene expression during development and differentiation. The association of chromatin with the nuclear periphery is cell type-specific, and has been implicated in gene regulation by dynamically modulating gene accessibility during normal development[11,16]. Nucleoplasmic proteins constitute a large component of the inner nucleus, but their role in chromatin remodeling during transcription and differentiation processes is poorly understood. Among the inner nuclear proteins, Matrin-3 (Matr3) is an abundant component and appears to be involved in multiple processes[17–19]. Matr3 interacts with other structural and regulatory proteins in the nucleus, controls RNA processing, and coding mutations cause rare genetic disorders[20–22]. A scaffolding role of Matr3 for regulatory proteins has been suggested in transcriptional control of pituitary cells[23]. Moreover, Matr3 expression in neural stem cells has been suggested to maintain the undifferentiated state, albeit limited to morphological observations[24]. To date, the extent, if any, to which Matr3 contributes to chromatin organization during transcription and differentiation remains unexplored. Very recently, Matr3 was identified as part of a protein complex that participates in X-chromosome silencing[25], demonstrating its regulatory potential at the chromatin level.

Here we have addressed whether Matr3 plays a critical role in chromatin structure and function. Our studies reveal unique aspects of the impact of nuclear protein-chromosomal organization on 3D genome structure and the molecular machinery underlying chromatin repositioning during development.

## Results

**Depletion of Matr3 leads to changes in nuclear architecture and accelerates erythroid maturation.** Blood cell development exemplifies a coordinated process that is accompanied by dramatic chromatin reorganization, thereby providing a model in which to interrogate chromatin dynamics during differentiation[26]. As a first step in assessing how the inner nuclear protein Matrin-3 (Matr3) impacts nuclear structure and gene expression, we deleted the entire gene body by CRISPR/Cas9 in mouse erythroleukemia (MEL) cells (Figs. 1a and S1a). Matr3 knockout (KO) cells proliferated at the same rate as parental cells (Fig. S1b), but were smaller in size and exhibited distinct cell morphology during DMSO-induced differentiation, suggestive of accelerated cell maturation (Fig. 1b). Consistent with this notion, erythroid-specific genes were expressed at a higher level in MEL Matr3 KO cells than in parental cells (Fig. 1c, d). To ensure that these findings were due to Matr3 loss rather than to inadvertent events during isolation of clones, we rescued the phenotype by reintroduction of full-length, expressible Matr3 cDNA (Figs. 1d and S1c). The consequences of Matr3 deletion were also determined in G1ER cells, another tractable model in which differentiation is conditional on activation of GATA-1[27]. Similar to MEL cells, globin gene expression was elevated in G1ER Matr3 KO clones (Fig. S1d).

To assess the global impact of Matr3 loss on erythroid cell maturation, we measured global RNA expression changes at early and differentiated stages. Erythroid-specific genes were expressed at a much higher level upon differentiation of Matr3 KO cells (Fig. 1e). Similarly, 533 previously reported erythroid genes[28] were also expressed at a higher level in KO cells (Fig. S1e). Differentiation is typically accompanied by specific changes in nuclear architecture. Using super-resolution microscopy, we observed that heterochromatin protein 1α (HP1α) was more dispersed and irregular in appearance, despite no appreciable change in expression in Matr3 KO cells (Figs. 1f, g and S1f). These findings suggest that Matr3 loss alters morphological boundaries of heterochromatin. Together, depletion of Matr3 resulted in accelerated erythroid cell maturation and distinct morphological changes in nuclear structure.

**Matr3 loss is accompanied by global chromatin reorganization.** Analysis of the interactions between different regions of chromatin identifies topologically associating domains (TADs) and classifies the genome into two compartments (A and B). We next explored global chromatin structure using a high-throughput chromosome conformation capture (Hi-C) assay. Although extensive chromosomal interaction patterns appeared largely unchanged (Fig. 2a, b), compartments and interactions in local regions were visibly altered on comparison of parental and Matr3 KO cells (Fig. 2c–e). To examine the global compartment changes, we measured the interactions between genomic loci aligned by values of the first principal component (PC1) from eigenvector decomposition[2]. In the saddle plots, the strengths of A and B compartments were quantified by calculating the ratio of homotypic (A-A or B-B) to heterotypic (A-B) compartmental interactions (Fig. 2f). Notably, the compartment strengths between the B compartments became stronger, while those between A-type domains were reduced in Matr3 KO cells, suggesting a requirement for Matr3 in maintenance of proper chromosome compartmentalization (Figs. 2f, g and S2a).

We next investigated the chromosomal domain boundaries that demarcate the dynamic 3D genomic structural unit, TAD. Domain boundaries were determined using the insulation profile of chromosomes[29], and we aggregated interaction data at the

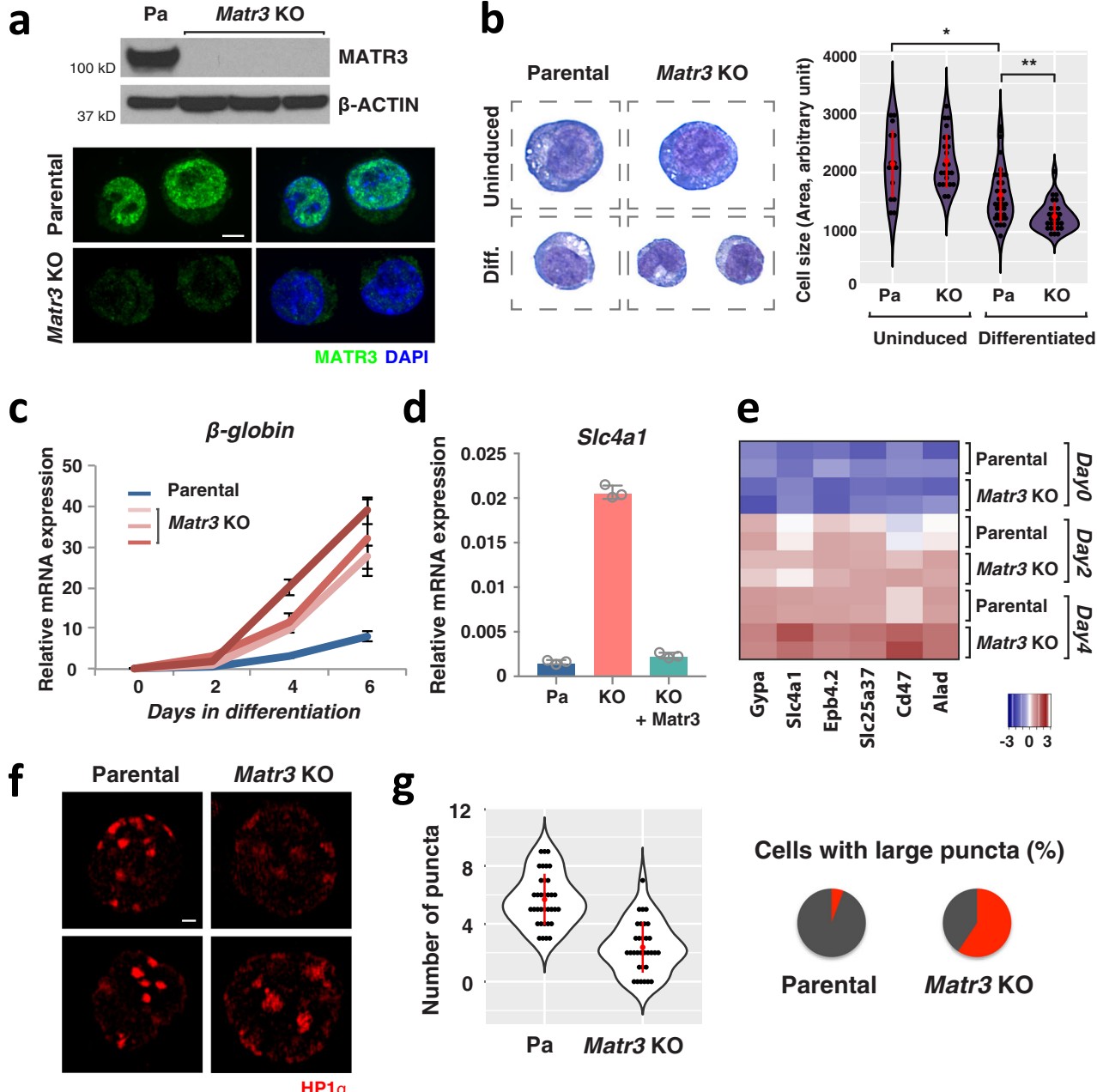

**Fig. 1 Altered nuclear structure and differentiation following Matr3 loss. a** MATR3 expression was assessed in parental and *Matr3* KO MEL cells by Western blot and immunohistochemistry. β-actin was used as a Western blot control. Scale bar, 5 μm. **b** May-Grunwald Giemsa staining was performed on uninduced and 5-day differentiated cells. During MEL cell differentiation, cell size is normally reduced ($p = 0.002$, by t test); the effect is more pronounced in *Matr3* KO cells ($p = 0.0004$, by two-sided t test). The number of examined cells was $n = 15$, $n = 23$, $n = 34$, and $n = 28$, respectively. **c** β-globin mRNA expression was greater in *Matr3* KO than normal MEL cells, indicative of enhanced cell differentiation ($n = 3$). Three independent *Matr3* KO clones by CRISPR/Cas9 were examined. **d** mRNA encoding *Slc4a1*, an erythroid-specific gene, was also increased in *Matr3* KO cells; expression of full-length *Matr3* cDNA (Fig. S1c) lowered *Slc4a1* expression to wild-type levels ($n = 3$). **e** Relative RNA levels based on z-score analysis demonstrated that expression of late erythroid genes was significantly higher in *Matr3* KO than wild-type cells upon differentiation. *P*-values obtained from the two-sided t test between two groups at each day were 1.94e-01, 1.37e-02, and 3.65e-07, respectively. **f** Heterochromatin was visualized by immunohistochemistry using anti-HP1α antibody and super-resolution microscopy. Scale bar, 1 μm. **g** The number of distinct puncta per cell ($p < 5.9e-11$; $n = 35$, $n = 32$, respectively) and percentage of cells with at least one large puncta were quantified (5.7% and 59.4%, respectively). A large puncta was defined as having the longest diameter that is more than twice the average. Data were the result of 2–3 independent experiments and error bars represent mean ± 1 s.d. Source data are provided as a Source Data file.

boundaries and compared the changes upon Matr3 loss. In *Matr3* KO cells, insulation at the boundaries was reduced, resulting in more interactions being observed across TAD boundaries (Fig. 2h). We also examined interactions within TADs to focus on local chromatin compaction by calculating the mean contact

frequency between the bins within a TAD. Curiously, intra-TAD contacts within compartment B were decreased, whereas interactions within compartment A increased, perhaps reflecting altered chromatin structure revealed by HP1α staining (Figs. S1g and 1f, g). Consistent with this hypothesis, analysis of TADs with

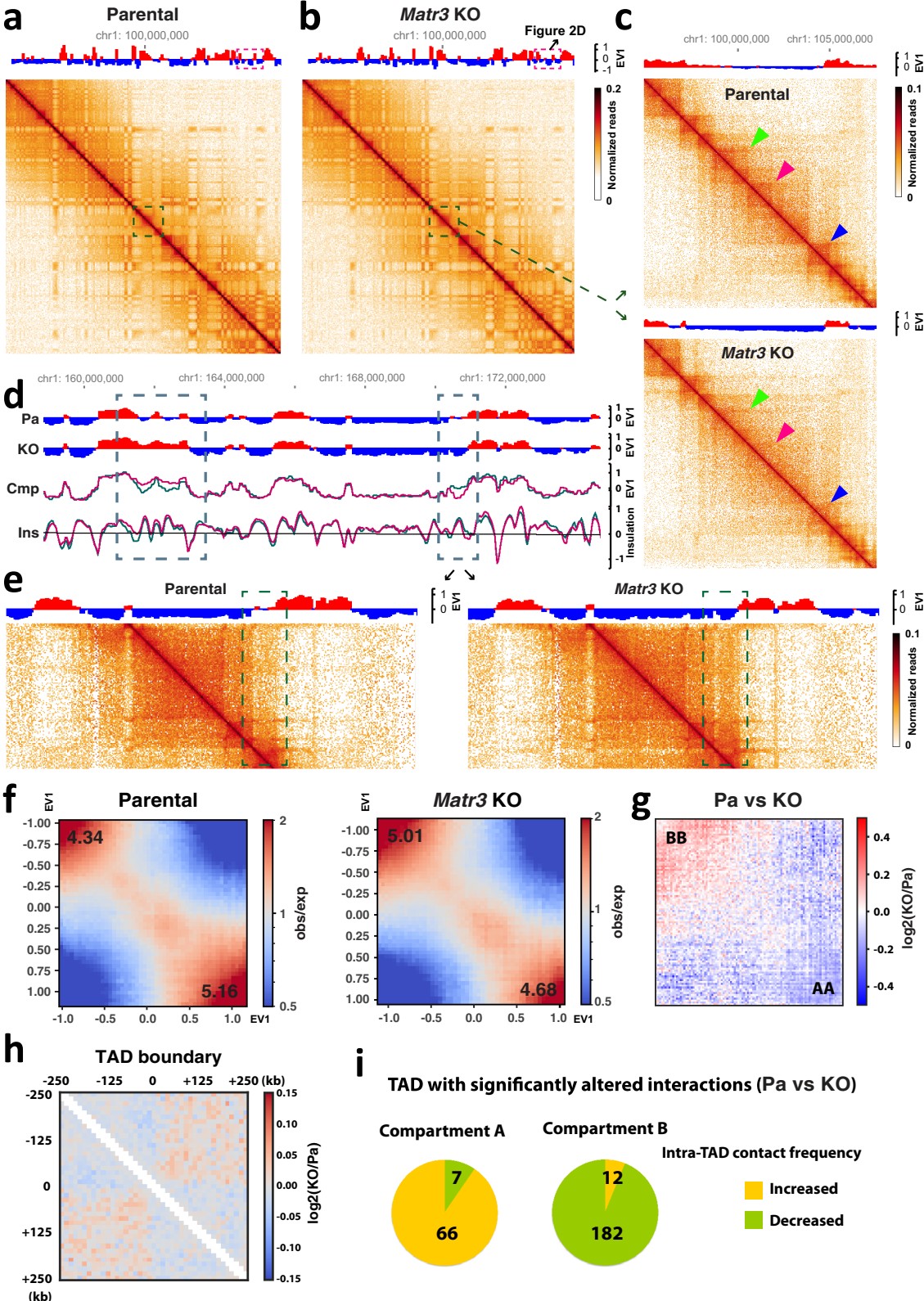

significantly altered interactions revealed that most of the TADs in A compartments, or A-type domains, exhibited increased intra-TAD Interaction frequency, whereas those in B-type domains displayed decreased contact frequency (Fig. 2i). In short, Matr3 loss was accompanied by a global reorganization of chromosomal structure.

**Chromatin structural alterations in absence of Matr3 resemble changes during differentiation.** Cell differentiation is accompanied by coordinated chromatin remodeling. Remarkably, we found that changes in chromatin contact strength during differentiation resemble those in cells lacking Matr3. During normal erythroid maturation, interaction strengths within B-type

**Fig. 2 Matr3 loss leads to global reorganization of 3D genome architecture. a, b** The first eigenvector (EV1) indicating the genomic compartment for chromosome (chr) 1 and Hi-C contact matrices at 500 kb resolution. **c** Representative region (green dashed box in Fig. 2a, b) of Hi-C data at 25 kb resolution with the compartment signal. Arrowheads indicate altered interactions. **d** Snapshot of Hi-C data at chr 1 (red dashed box in Fig. 2a, b) showing each compartment, compartment difference, and insulation score of parental (green) and *Matr3* KO (magenta) cells. Compartment switch is apparent in the indicated regions marked with dashed boxes. One of the altered regions is plotted with Hi-C contact matrices at 25 kb resolution in (**e**). **f** Contact frequency enrichment for compartments A (bottom right) and B (top left). Compartmentalization saddle plots were calculated by normalized interaction frequencies between loci of 100 kb bins arranged by their eigenvector values (EV1). The numbers of the heatmaps indicate the average compartment strength quantified by calculating the ratio of homotypic (A-A or B-B) to heterotypic (A-B) compartment interactions of the top 20% sorted EV1 values. The difference was calculated as log2 ratio of average interaction intensity (obs/exp) in *Matr3* KO and control at 50 kb, and shown as a differential map **g**, and the change was further confirmed in Fig. S2a. **h** Hi-C interaction data binned at 25 kb resolution was aggregated at TAD boundaries and the difference was calculated by log2 ratio of *Matr3* KO and parental cells. Results of replicate experiments for **f–h** are shown in Fig. S1h-h'. **i** Analysis of the dynamically altered TADs in *Matr3* KO revealed that most of the TADs in compartment A had increased intra-TAD interaction frequency, whereas those in B had decreased contact frequency. Numbers in the pie chart represent TADs with significantly changed intra-TAD interaction frequency, which was defined as $p < 0.05$ and log2 fold change > 0.15 or < -0.15.

domains increased, whereas contacts within A compartments became weaker (Figs. 2f, 3a–b, and S2b). The frequency of interactions within TADs decreased in compartment B, and the majority of TADs that were significantly altered in compartment B exhibited decreased intra-TAD contact frequency during differentiation, similar to be seen in the absence of Matr3 (Figs. 3c and S2g). Indeed, TADs with significantly altered interactions in *Matr3* KO tended to have significantly altered contact frequencies during differentiation (Fig. S2c). This pattern was also observed at the domain boundary, reflected by weaker insulation at the boundaries and more interactions across TAD boundaries upon differentiation (Fig. 3d). Consistent with these findings, the average insulation score[29] of TAD boundaries increased, indicating weak insulation during differentiation, and similarly in *Matr3* KO cells (Fig. 3e). In fact, the global reorganization of chromosomal interactions during differentiation appeared to be accelerated in the absence of Matr3 (Figs. 3f, g and S2i, j). The size of TADs, identified using two analytical methods[30,31], tended to increase in *Matr3* KO cells (Figs. 2c, 3g, and S2j), similar to that observed during differentiation, with a corresponding decrease in the overall number of TADs (Fig. S2i, j).

To access the genomic features of chromatin regions at a higher resolution, we performed the assay for transposase-accessible chromatin with high throughput sequencing (ATAC-seq) that identifies accessible chromatin regions. Notably, the newly opened regions in *Matr3* KO, as compared to parental, cells were enriched for GATA motifs, which provide the binding sites for the master hematopoietic transcription factor GATA-1 (Fig. 3h). These cis elements are generally more accessible in differentiated cells, suggesting that loss of Matr3 may increase the probability of binding of critical developmental regulators to chromatin. Regulation of gene expression requires coordinated interactions of transcriptional activators with promoters and transcription start site (TSS)-distal regulatory elements. We therefore assessed the relative localization of open chromatin regions to TSS and enhancers in parental and *Matr3* KO cells (Fig. 3i, j). The number of ATAC-seq peaks assigned to distal enhancers was greater in the *Matr3* KO cells, whereas the number of peaks was similar in proximal regions. Thus, chromatin of *Matr3* KO cells resembles that of more differentiated cells. More specifically, distal regulatory regions associated with cell maturation become more accessible upon loss of Matr3.

**Matr3 interacts with architectural proteins (CTCF and cohesin) and affects their chromatin occupancy.** Emerging studies of genome structure indicate that architectural proteins function cooperatively to organize chromatin[32,33]. To identify protein interaction partners of Matr3, we employed affinity purification

of biotinylated Matr3 in MEL cells, followed by mass spectrometry (Fig. 4a). Cells for this analysis were generated by functional rescue of *Matr3* KO cells with a FlagBio-tagged *Matr3* cDNA. As anticipated for an abundant inner nuclear component, mass spectrometry identified numerous proteins, including those involved in RNA processing, chromatin remodeling, and transcription (Fig. S3a). Previous studies of Matr3 have focused mainly on its RNA binding properties and proposed role in regulating alternative splicing[21,34]. To investigate the extent to which isoforms differentially regulated by Matr3 affect gene regulation, we compared alternative splicing events to gene expression changes using RNA-seq data (Fig. S3b). Only a subset of alternative splicing events was associated with the transcriptome shift of *Matr3* KO, suggesting that other factors contribute to altered gene expression. In fact, we found that Matr3 also interacts with several proteins involved in chromatin remodeling, such as Cbx3 (heterochromatin protein 1γ), Esco2 (cohesin acetyltransferase), CTCF, and Rcc1 (regulator of chromosome condensation). In particular, CTCF has been reported to interact with Matr3[35]. The role of CTCF and cohesin complexes in chromatin conformation and their contribution to gene regulation has been well characterized in recent studies[36,37]. Consistent with proteomic data pointing to interactions between Matr3 and CTCF/cohesin, we observed that Matr3 coimmunoprecipitates with these proteins (Fig. 4b). In addition, affinity purification of the cohesin complex with Smc1a antibody in human acute myeloid leukemia (AML) cells, followed by mass spectrometry, identified Matr3 (Fig. 4c left). The abundance of Matr3 was depleted when the cohesin complex was purified with Smc1a antibody in Stag2-mutant AML cells, supporting our findings on the interaction of the cohesin complex with Matr3 (Fig. 4c right).

We then asked whether Matr3 loss alters chromatin occupancy of its interacting partners by performing ChIP-seq for CTCF and the core cohesin component Rad21. Expression levels and genome-wide distribution of CTCF and cohesin remained largely unchanged between parental and *Matr3* KO cells (Fig. 4d and S3c). However, quantitative comparison after normalization using rescaling of ChIP-seq signals by common peaks[38] indicated that a greater number of chromatin sites were occupied by CTCF and cohesin in parental as compared with *Matr3* KO cells (Fig. 4d). Similarly, upon analysis of statistically significant and differentially bound regions measured by the difference in read density of ChIP-seq[39], a greater number of differentially bound CTCF and cohesin sites were identified in parental compared to *Matr3* KO cells (Fig. 4e). The probability of genome-wide contact calculated from Hi-C data can determine the linear density of cohesin on chromatin[40,41]. Consistent with the results from ChIP-seq analysis, *Matr3* KO cells tended to have a flatter minimum of

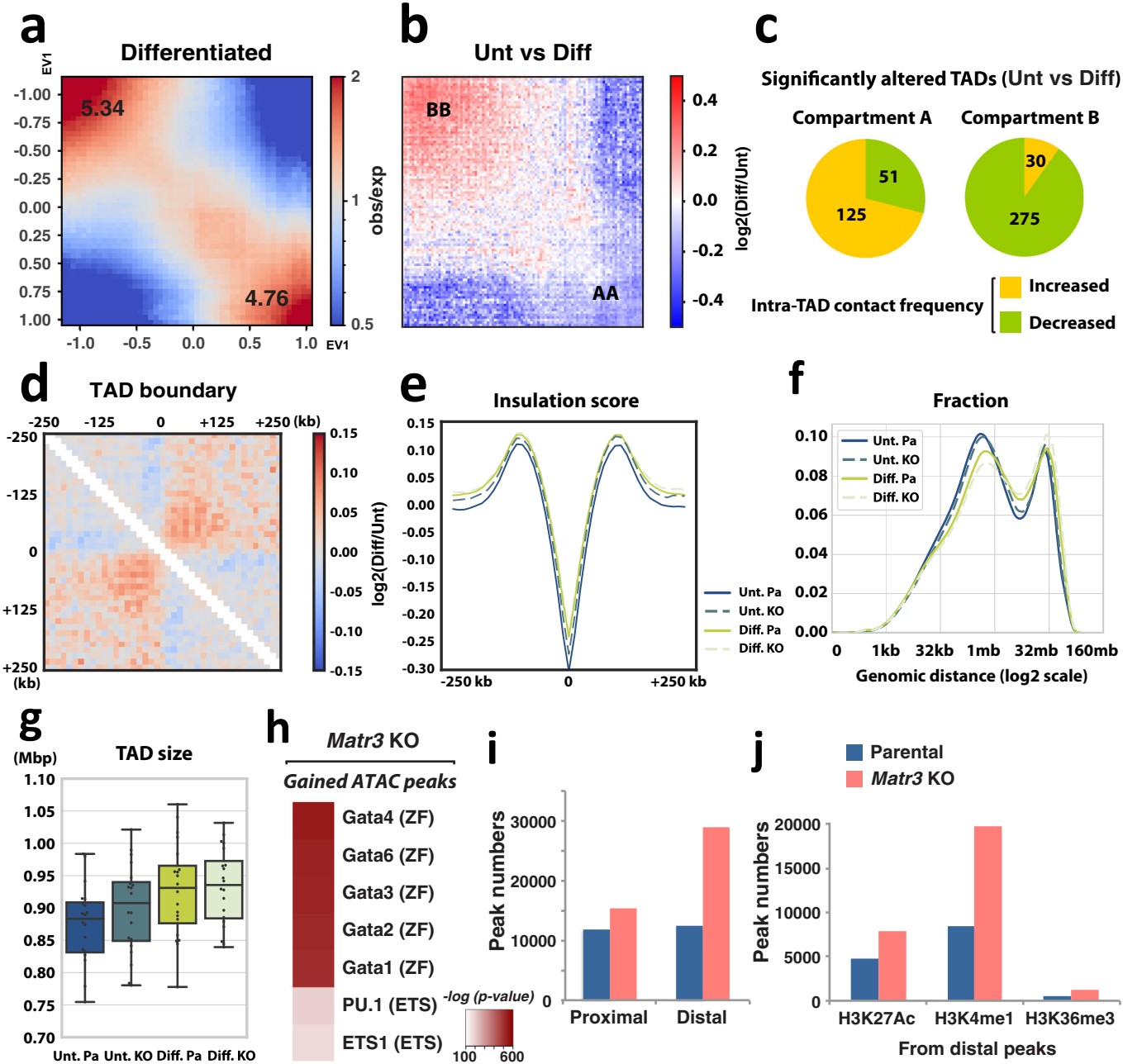

**Fig. 3 Alterations of chromatin structure during differentiation resembled that in *Matr3* KO cells, and Matr3 loss opens regulatory chromatin regions specific to differentiation. a** Contact frequency enrichment for compartment A (bottom right) and B (top left) in the differentiated parental cell. The difference between uninduced (Unt) and differentiated (Diff) cells was calculated as the log2 ratio of average interaction intensity (obs/exp) at 50 kb, and shown as a differential map **b** and the change was further confirmed in Fig. S2b. Results of replicate experiments are shown in Fig. S2d-d'. **c** Analysis of the significantly altered TADs revealed that most of the TADs during differentiation in compartment A had increased intra-TAD interaction frequency, while those in B had decreased contact frequency. Numbers in the pie chart represent TADs with significantly changed intra-TAD interaction frequency. **d** Hi-C interaction data binned at 25 kb resolution was aggregated at TAD boundaries and the difference was calculated by log2 ratio of differentiated and uninduced cells. Results of replicate experiments are shown in Fig. S2e-e'. **e** Average insulation score[29] across TAD boundaries increased during differentiation and in *Matr3* KO. Results of replicate experiments are shown in Fig. S2f-f'. **f** Genomic length distribution of Hi-C contacts. Results of replicate experiments are shown in Fig. S2h-h'. **g** Average TAD size for each chromosome identified using[30] across two independent experiments (Unt.Pa vs. Unt.KO: $p = 0.007$, Unt.Pa vs. Diff.Pa: $p = 0.0004$, Diff.Pa vs. Diff.KO: $p = 0.46$, by two-sided Wilcoxon signed-rank test, respectively; Cohen's $d = 0.44$, Cohen's $d = 0.84$, Cohen's $d = 0.08$, respectively). Center lines, boxes, and whiskers represent the median value, first and third quartiles, and 1.5 interquartile range of the samples, respectively. **h** ATAC-seq peaks unique to *Matr3* KO compared to parental cells (gained peaks) were used for motif analysis. The natural logarithm of the p-values calculated using the binomial distribution are shown in the heatmap and in Table S1. **i** The number of ATAC-seq peaks proximal or distal to TSS was counted. Then, the peaks in the distal region were further analyzed to overlap the ChIP-seq peaks for histones **j**.

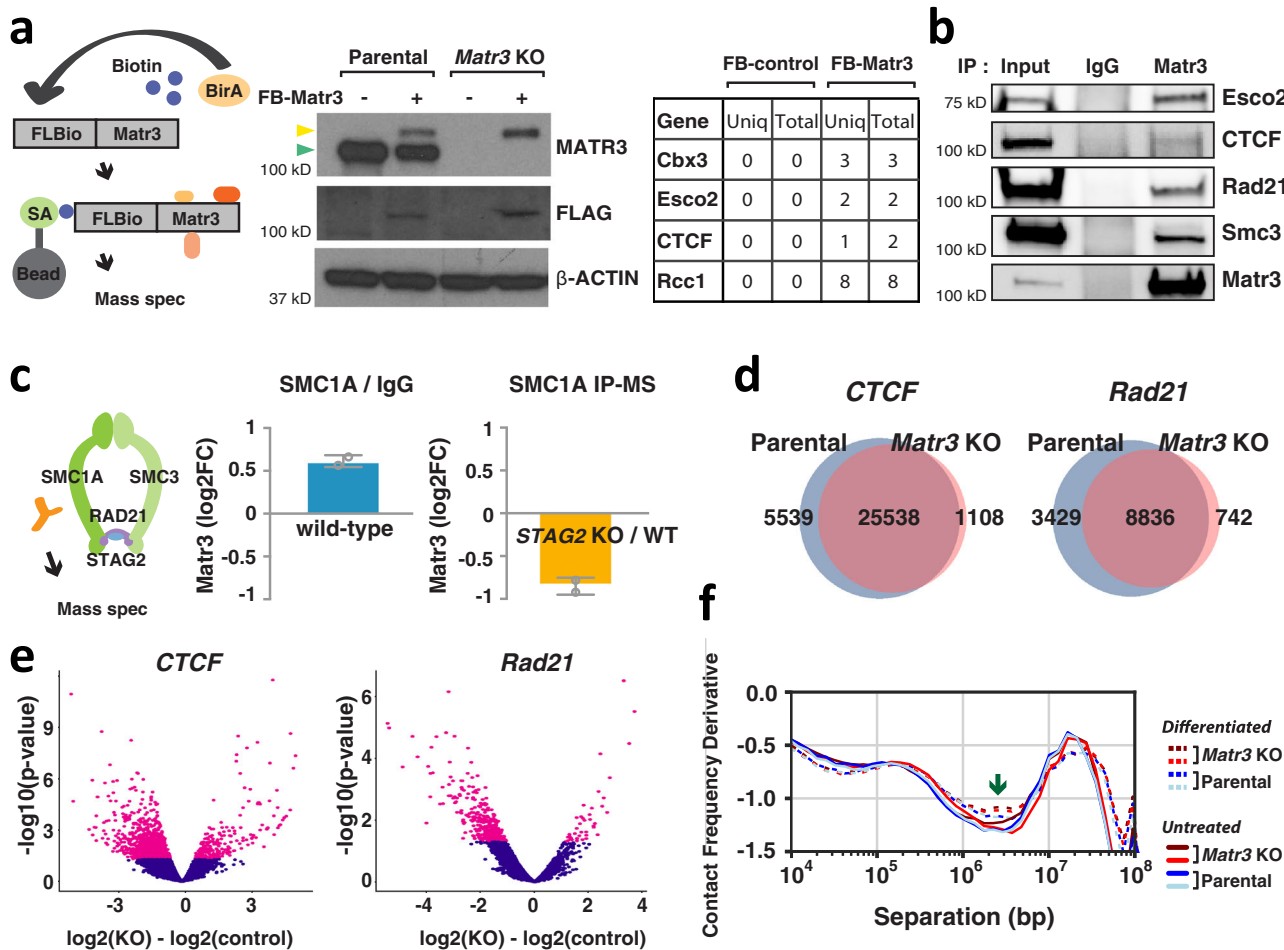

**Fig. 4 Matr3 interacts with architectural proteins including CTCF and cohesin, and Matr3 loss alters their chromatin occupancy. a** Parental and *Matr3* KO MEL cells were engineered to express BirA and Flag-Biotin tagged Matr3. Western blot discriminated endogenous (green arrow) and/or biotinylated (yellow arrow) forms of Matr3 protein. Biotinylated form of Matr3 was recovered using streptavidin magnetic beads. BirA expressing cell was used as a control. From the result of mass-spectroscopy (MS) analysis (Fig. S3a), the number of unique and total peptides of the selected chromatin remodeling factors are shown in the table. Data were the result of 2 independent experiments. **b** Endogenous Matr3 protein was immunoprecipitated and its interaction with other proteins was examined by Western blot. Data were the result of 2–3 independent experiments. **c** The Smc1a subunit of the cohesin complex was pulled down in *Stag2* wild type (WT) and KO U937 cells and subjected to mass-spectroscopy (MS)[80]. (left) Matr3 interaction to the cohesin complex is revealed by log2FC of Smc1a and IgG immunoprecipitation (IP)-MS in *Stag2* WT cells. (right) The disrupted Matr3 interaction on the cohesin complex in *Stag2* KO is measured by log2FC of Smc1a IP-MS in *Stag2* KO and WT cells. Error bars represent mean ± 1 s.d. across two independent experiments. **d** ChIP-seq datasets of CTCF and Rad21 in parental and *Matr3* KO cells were quantitatively compared[38]. The results of at least two independent experiments were combined to generate a more stringent peak list. The numbers of specific and shared peaks in each comparison are shown in the Venn diagram. **e** Differentially bound CTCF and Rad21 sites between parental and *Matr3* KO cells were identified using[39]. Sites with *p*-values from the Wald test less than or equal to 0.05 are shown in red on the plot. **f** Genome-wide contact probability calculated from Hi-C data (Fig. S2k). Derivative plot of the contact probability curve suggests low cohesin density in *Matr3* KO compared to parental cells. The flatter minimum expected to be seen with a reduced level of cohesin occupancy is indicated by an arrow.

contact frequency derivative, which is expected to be seen with reduced cohesin occupancy (Figs. 4f and S2k).

**Matr3 loss alters chromatin contacts to the nuclear structure**. In *Matr3* KO MEL cells, both CTCF and cohesin binding to chromatin was markedly reduced at a subset of genomic regions. Global changes in gene expression, as assessed by RNA-seq analysis, were unremarkable in *Matr3* KO cells; however, changes became more evident upon cell differentiation (Fig. S3d, e). Of the genes displaying reduced CTCF and Rad21 occupancy in their vicinity, Mbd1 was most significantly down-regulated in expression in the absence of Matr3, and therefore was chosen for detailed study (Figs. 5a, b and S3f). We first asked whether Matr3

might be directly involved in the regulation of this locus by performing ChIP-seq and CUT&RUN[42] with available Matr3 antibodies. We failed to identify a substantial numbers of peaks perhaps due to weak affinity of the antibody or lack of proximity of Matr3 to DNA. To circumvent this problem and ask whether Matr3 resides near the *Mbd1* gene at the position occupied by CTCF and cohesin, we adopted a chromatin CAPTURE procedure[43]. This method employs sequence-specific single guide RNA (sgRNA) to direct biotinylated dCas9 protein to a region of interest for subsequent streptavidin affinity purification that allows isolation of the targeted chromatin and associated protein complexes[43,44]. We designed sgRNAs that bind to the putative cis-regulatory element bound by CTCF and cohesin near *Mbd1*, and sgRNAs that target a nearby upstream, as a control

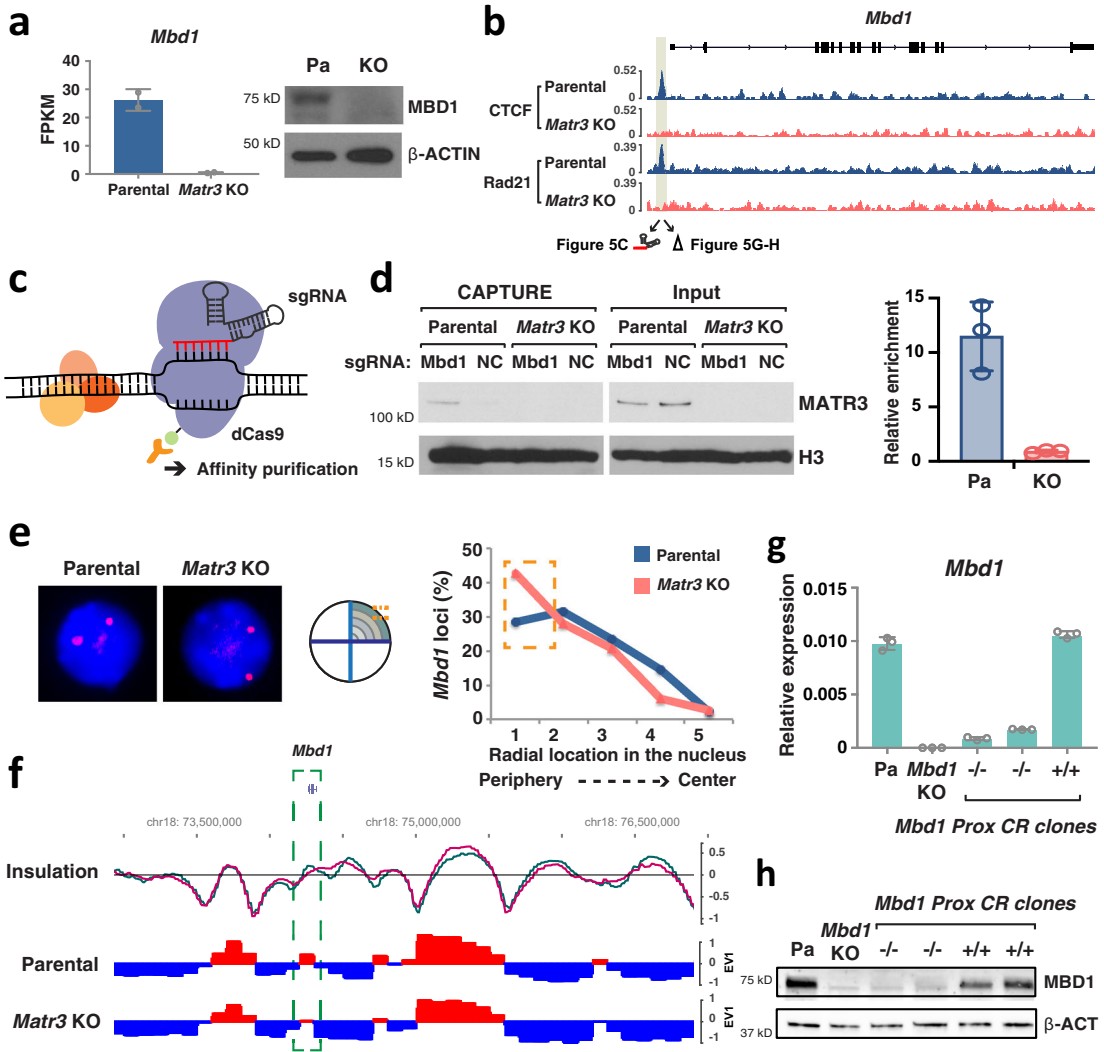

**Fig. 5 Matr3 loss alters chromatin contacts to the nuclear structure. a** Reduced expression level of Mbd1 was revealed by RNA-seq from two independent experiments and Western blot. β-actin was used for a Western blotting control. **b** Chromatin occupancy of CTCF and Rad21 in parental and *Matr3* KO cells near *Mbd1* locus is shown. **c** CAPTURE experiment[43] was performed using sgRNAs directed to a putative cis-regulatory element near *Mbd1* which is occupied by CTCF and cohesin, and an adjacent upstream, as a negative control (NC) (Figs. 5b and S4b). **d** Purification of biotinylated dCas9 and Western blot indicated that Matr3 interacted with the putative element (Mbd1) but not with the upstream control (NC). Relative enrichment of Matr3 protein at the putative element to the upstream control was quantified from three independent experiments. **e** Radial location of the *Mbd1* gene was observed by 3D DNA FISH. For quantification, the nuclear radius was divided into five shells and the number of signals in each shell was counted. The graph is the result of 2–3 independent experiments and the change was significant under a two-sided *t* test ($p = 0.006$; Cohen's $d = 0.32$). **f** Snapshot of Hi-C data at the *Mbd1* locus showing compartments in parental and *Matr3* KO cells and the difference in insulation score (parental in green and *Matr3* KO in magenta). **g**, **h** Deletion of CTCF/cohesin binding region (shaded box in Fig. 5b) by CRISPR/Cas9 reduced Mbd1 expression. RNA (**g**) and protein (**h**) levels were examined by RT-PCR ($n = 3$) and Western blotting, and compared to *Mbd1* KO cells. β-actin was used for a Western blot control. Data were the result of 2–3 independent experiments and error bars represent mean ± s.d. Source data are provided as a Source Data file.

(Figs. 5c and S4b–d). Affinity purification of biotinylated dCas9 targeted to the CTCF/Rad21-bound regulatory element, but not the control region, revealed significant enrichment of Matr3 protein (Fig. 5d). This finding, together with the absence of CTCF/cohesin occupancy and diminished expression of Mbd1 in *Matr3* KO cells, strongly suggests that the presence of Matr3 at this region is required for the chromatin association of CTCF and cohesin to coordinate *Mbd1* transcription.

To detect the relative position of the *Mbd1* gene locus inside the nucleus, we employed fluorescence in situ hybridization (FISH) (Fig. 5e). In *Matr3* KO cells, *Mbd1* loci appeared to be more frequently situated closer to the nuclear periphery, which is

typical for genes within inactive chromatin. Consistent with this, the active A compartment containing the *Mbd1* gene body was weakened in the global interaction analysis by Hi-C (Fig. 5f). CTCF/cohesin-mediated chromatin organization is required for proper regulation of gene expression[9,10]. In fact, CTCF/cohesin-occupied region upstream of *Mbd1* was located near a loop anchor, and the insulation became weaker in *Matr3* KO cells (Fig. S4a). Therefore, we examined the region where CTCF/cohesin occupancy was perturbed in *Matr3* KO cells (Fig. 5b). Using CRISPR/Cas9-mediated deletion, we generated cells with deletion of the entire *Mbd1* gene and cells in which the CTCF/cohesin binding region near *Mbd1* was specifically removed.

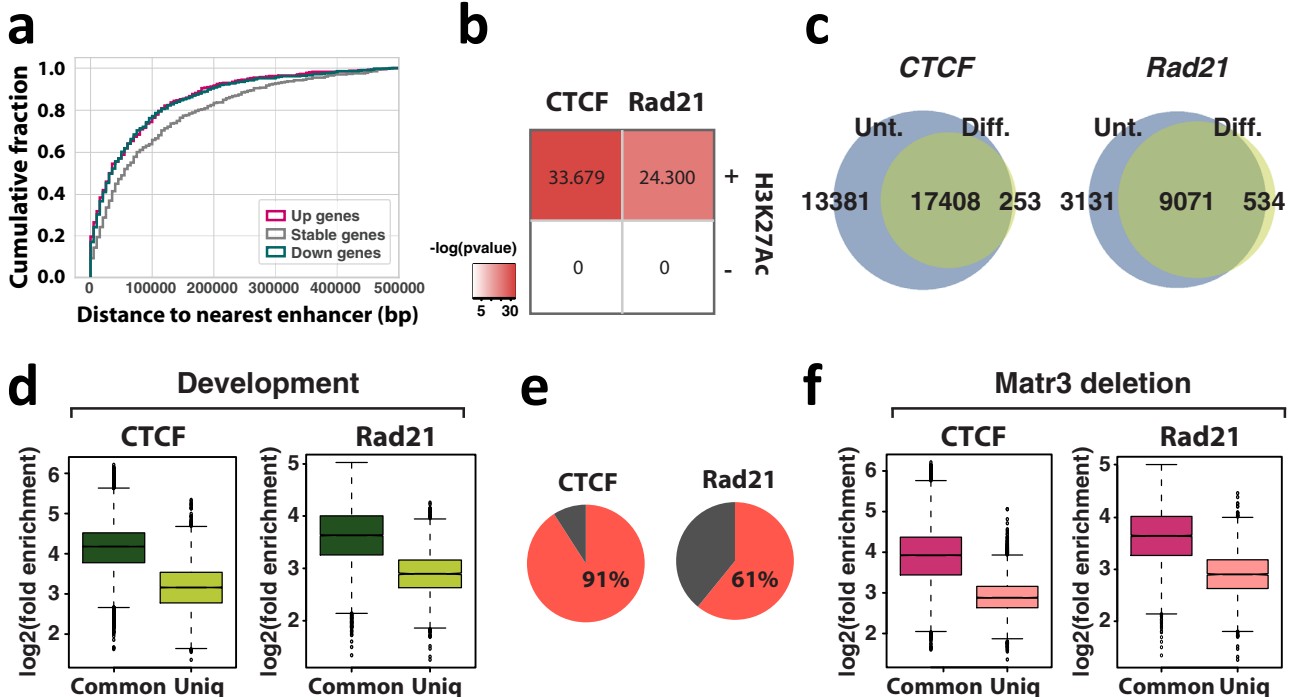

**Fig. 6 Low occupancy sites of CTCF and cohesin are susceptible to developmental regulation and Matr3 loss. a** Cumulative probability distribution of the distance to the nearest enhancer for up- and down-regulated genes versus stable genes in *Matr3* KO compared to parental cells. Genes whose expression was altered upon Matr3 deletion resided at shorter genomic distances under a two-sided Kolmogorov-Smirnov (KS) test (up vs. stable: $p = 3.62$e-06, down vs. stable: $p = 1.93$e-09, up vs. down: $p = 0.495$, respectively). **b** Differently occupied regions of CTCF and Rad21 in *Matr3* KO compared to parental cells nearby H3K27Ac marked chromatin associate with genes whose expression was altered upon Matr3 loss during differentiation. Significance of enrichment was calculated using the one-sided Fisher's exact test. The odds ratios for significant enrichment for CTCF and Rad21 were 2.01 (95% confidence interval (c.i.) = 1.7–2.4) and 2.03 (95% c.i. = 1.7–2.5), respectively. **c** ChIP-seq data sets of CTCF and Rad21 from uninduced (unt) and differentiated (diff) cells were quantitatively compared[38], and the results of at least two independent experiments were combined to generate a more stringent peak list. **d** After classification of regions into altered (uniq) and maintained (common) during differentiation (Fig. 6c), the enrichment of the peaks in each experiment was investigated using[86] ($p < 2.2$e-16, $p < 2.2$e-16, respectively, by two-sided t test; Cohen's $d = 1.48$, Cohen's $d = 1.24$, respectively). The results were confirmed in 2-3 independent experiments (Fig. S5a-c). **e** Percentage of sites with reduced CTCF and Rad21 binding during differentiation in the binding regions lost in *Matr3* KO compared to parental cells. **f** After classification of binding regions into maintained (common) and altered (uniq) upon Matr3 loss (Fig. 4d), the enrichment of the peaks in each experiment was measured ($p < 2.2$e-16, $p < 2.2$e-16, respectively, by two-sided t test; Cohen's $d = 1.22$, Cohen's $d = 1.26$, respectively). The results were confirmed in 2-3 independent experiments (Fig. S5d-f). In box plots, the centre line represents the median, box limits show upper and lower quartiles, and whiskers extend to 1.5 × interquartile range.

Notably, removal of this binding site reduced Mbd1 expression to a level similar to that in *Mbd1* KO cells, providing functional validation of the relevance of the CTCF-cohesin-Matr3-associated regulatory element (Fig. 5g, h). We cannot exclude, however, that the cis element we have removed binds other factors critical for Mbd1 expression. Taken together, these findings indicate that binding of CTCF and cohesin to select chromatin regions is dependent on Matr3 and necessary for proper chromatin interactions and gene expression.

**Sites of low CTCF and cohesin occupancy correlate with developmental regulation and sensitivity to Matr3 loss.** Despite significant changes in chromatin architecture, the consequences of Matr3 loss on gene expression were remarkedly limited in undifferentiated cells, but enhanced upon differentiation (Figs. 1e, 2, and S3d-e). Thus, chromatin changes associated with Matr3 loss may facilitate transcriptional responses to developmental cues. Indeed, recent studies have shown that many signal-response enhancers, such as development-specific elements, are in contact with target promoters prior to signal transduction[45–47]. These pre-existing enhancer-promoter loops presumably facilitate rapid transcriptional activation. Moreover, we found that

expression of genes close to enhancers was affected by Matr3 loss (Fig. 6a), suggesting that the local looping was perturbed.

Architectural proteins, such as CTCF and cohesin, establish the boundaries between TADs and mediate interactions between regulatory elements within chromatin domains. Changes in their chromatin occupancy occur during differentiation, and also appear to be associated with dynamic chromatin contacts that regulate inducible gene expression[46,48,49]. For example, lineage-specific loops in epidermal precursor cells that provide a framework for enhancer contacts to the differentiation-related genes prior to terminal differentiation are established by cohesin occupancy that is not present in pluripotent cells[46]. Therefore, we investigated those regions that displayed changes in CTCF and cohesin occupancy upon Matr3 loss in relation to differentiation (Fig. 4d). Notably, CTCF and Rad21 sites with altered occupancy near enhancers were strongly associated with altered gene expression during differentiation (Fig. 6b), suggesting that they perturb chromatin contacts that provide a regulatory infrastructure for the transcription of erythroid specific genes.

ChIP-seq experiments revealed that CTCF and Rad21 maintain occupancy at many chromatin regions during differentiation (Fig. 6c). However, quantitative comparison of ChIP-seq data and the probability of genome-wide contacts calculated from Hi-C

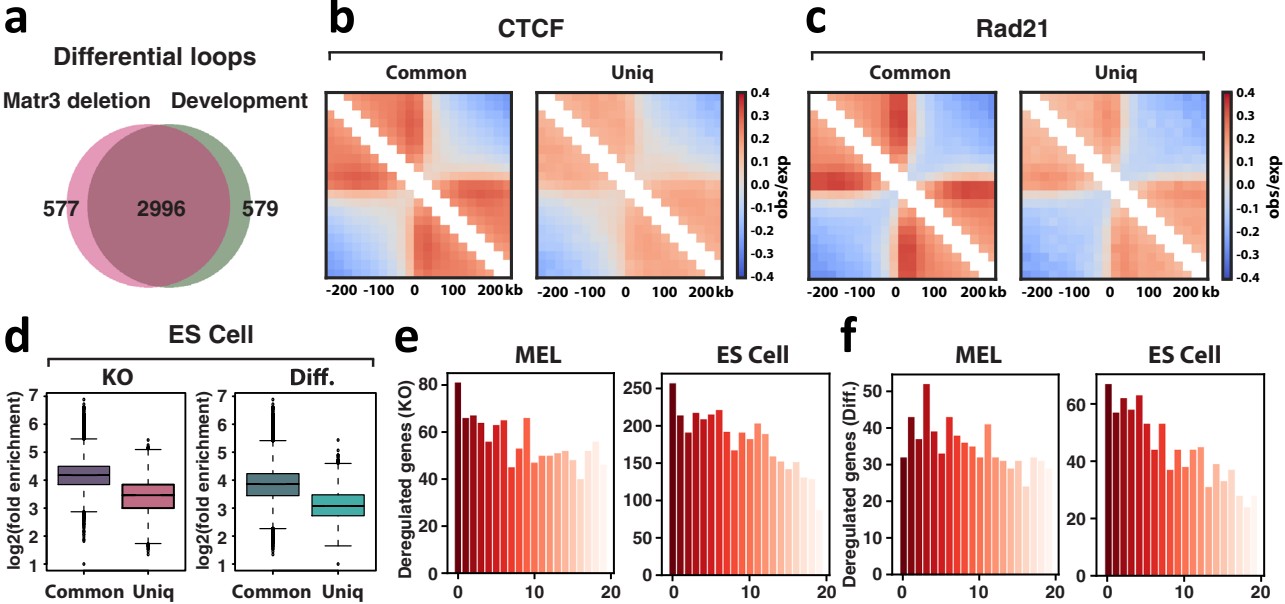

**Fig. 7 Impact of Matr3 loss on chromosomal architecture extends to embryonic stem (ES) cells. a** 3573 and 3575 differential loops lost in *Matr3* KO and during differentiation, respectively, were identified and significantly overlapped (2996, 84%). Random overlap for lost differential loops: 212 (out of 3650), 6%, *p* = 1e-5 from the permutation test. **b, c** Interaction pile-up maps of Hi-C data at the boundaries defined by each set of CTCF **b** and Rad21 **c** ChIP-seq peaks. The normalized pile-up interaction frequency observed over expected values indicates stronger insulation at regions of parental cells that are maintained (common) than at altered sites (uniq) in *Matr3* KO cells. The results were confirmed in 2-3 independent experiments (Fig. S5i-j). Hi-C data of *Matr3* KO compared to parental cells at altered sites are shown in Fig. S6a-b'. **d** ChIP-seq data sets of CTCF in mouse ES cells were quantitatively compared, and the results of independent experiments were combined to generate a more stringent peak list as described above (Fig. S6g). Then, the enrichment of the peaks in each experiment was investigated and confirmed (Fig. S6i). CTCF enriched less strongly at altered sites (uniq) than sites maintained (common) in *Matr3* KO cells and during differentiation (diff) (*p* < 2.2e-16, *p* < 2.2e-16, respectively, by two-sided t test; Cohen's *d* = 1.31, Cohen's *d* = 0.97, respectively). In box plots, the centre line represents the median, box limits show upper and lower quartiles, and whiskers extend to 1.5 × interquartile range. The CTCF peak-mapped genes were placed into ventiles (x-axis) from highest to lowest gene scores corresponding to genes with the most significant peak changes. The genes within each ventile were then overlapped with genes whose expression was altered (y-axis) in *Matr3* KO compared to parental cells **e** and during differentiation **f**. Analysis was performed on both MEL and mouse ES cells and the significance of the association was assessed using the one-sided hypergeometric test (*p* = 9.5e-08, *p* = 2.3e-26, respectively in **e** and *p* = 6.6e-04, *p* = 4.4e-14, respectively in **f**). Fold overlap over random: 1.4503, 1.428, 1.4857, 1.514, respectively.

data suggested that CTCF/cohesin occupancy tends to decrease upon MEL cell differentiation (Figs. 6c and 4f). Moreover, we found that CTCF and cohesin in parental cells were already weakly bound to regions at which occupancy was reduced on differentiation (Figs. 6d and S5a–c). Recent reports indicate that variable binding regions for CTCF and cohesin between cell types tend to be weak binding sites[50–52]. Hence, we reasoned that sites of low CTCF and cohesin occupancy might be more sensitive to changes in the local cellular environment, such as interacting scaffold proteins like Matr3. To test this model, we compared the variable chromatin regions in the absence of Matr3 to the sites that change during differentiation. Indeed, the majority of sites with reduced CTCF and Rad21 binding during differentiation were lost in the absence of Matr3 (Fig. 6e). We confirmed that CTCF and Rad21 in parental cells were also weakly bound to sites that were lost upon Matr3 loss (Figs. 6f and S5d–f). Consistent with the correlated changes in CTCF and Rad21 upon Matr3 loss and during differentiation, chromatin loops altered by loss of Matr3 correlate significantly with loops changed during differentiation (Figs. 7a and S5g). In fact, Hi-C interaction data at the boundaries defined by ChIP-seq peaks indicated that the occupancy of CTCF and Rad21 correlated with the level of insulation between neighboring topological domains. Chromatin insulation was reduced at the boundaries containing weak CTCF and Rad21 sites that were lost in the absence of Matr3, and more interactions were observed across the domain boundaries (Figs. 7b, c and S5i, j). A high proportion of sites with reduced

CTCF/cohesin occupancy resided within compartment B, suggesting that reduced chromatin binding of architectural proteins plays a role in reducing chromatin interaction frequency between B domains within TADs (Fig. S5h).

**Impact of Matr3 loss on chromosomal architecture extends to embryonic stem (ES) cells.** Our findings regarding the relationship between chromatin architectural proteins CTCF/cohesin and Matr3 emanate from analysis of a convenient model of red cell differentiation, and therefore raise the question of whether they are unique to this cell context or of more general relevance. To address this critical issue directly, we assessed chromatin occupancy of CTCF and cohesin (Rad21) in mouse embryonic stem (ES) cells of both parental and *Matr3* KO genotypes, and specifically interrogated changes in occupancy and gene expression upon differentiation accompanying removal of LIF. Similar to MEL cells, *Matr3* KO ES cells proliferated at the same rate as parental cells, and changes in gene expression were more evident upon differentiation (Fig. S6c–e). We observed that genes whose expression increased in *Matr3* KO ES cells during differentiation were enriched in gene sets implicated in cell differentiation and development (Fig. S6f). As in MEL cells, *Matr3* KO ES cells displayed a reduced number of high confidence CTCF and Rad21 binding sites (Fig. S6g). Moreover, regions with reduced occupancy of CTCF and Rad21 upon *Matr3* KO or differentiation contained weaker peaks in parental cells (Figs. 7d and S6h, i). Furthermore, as in MEL cells, reduced binding of CTCF was

strongly associated with gene expression changes in *Matr3* KO cells, as assessed from global RNA-seq (Fig. 7e). Most notable was the strong positive correlation between the most variable CTCF binding sites upon *Matr3* KO and differential gene expression during differentiation in both MEL and ES cell contexts (Fig. 7f). We conclude, therefore, that the impact of Matr3 on chromosomal architecture and developmental gene expression is not limited to a single cell type, but instead reflects a conserved feature of the interaction of an inner nuclear protein and chromatin.

## Discussion

**Nuclear architecture contributes to chromosome compartmentalization and organization and influences cell differentiation**. Active and inactive regions of chromatin, corresponding to the A and B compartments, respectively, are organized and separated spatially in the nucleus. Though much remains to be understood regarding how these features are formed and maintained, recent studies implicate phase separation of intrinsically disordered regions of proteins as a contributor to the compartmentalization of euchromatin and heterochromatin[3–7,53]. For example, HP1α droplets induce the formation of heterochromatic microphases, whereas clusters of transcriptional regulators form euchromatic condensates. Here, we identified increased interaction strength within the B compartments and decreased strength between the A-type compartments following loss of Matr3, a finding that resembles changes seen upon disruption of scaffolding nuclear speckles[54]. Thus, nuclear structural proteins appear to play a role in the organization of chromatin compartmentalization. By super-resolution microscopic examination of HP1α, we found that Matr3 loss affects chromatin boundaries. As Matr3 has intrinsically disordered regions[55], we speculate that it may also contribute to spatial separation of the genome through formation of liquid condensates.

Chromosomal organization is critical for proper gene expression and differentiation. Extensive changes in chromatin interactions and compartmentalization were observed during neural development with decreased TAD numbers, as well as reduced interaction strength between the A compartments and increased strength between the B-type domains;[8] yet whether the nuclear structure had an effect remained unexplored. Following Matr3 loss, chromatin architecture was perturbed and resembled that evident in a more differentiated state at both compartment and TAD levels. Similarly, chromatin reorganization during differentiation was accelerated in *Matr3* KO cells, demonstrating its role in maintaining chromatin structure in dynamic conditions. Perhaps due to cell type-specific 3D chromatin architecture[56,57], the set of altered genes varies between cell types. For example, expression of Mbd1 was markedly affected by Matr3 loss in MEL cells, but not changed in ES cells. Unlike pronounced chromatin remodeling, gene expression changes were less remarkable in *Matr3* KO cells, but became more evident during differentiation in both cell types. Local interactions between cis regulatory elements are extensively reorganized during differentiation, and mediated by the combinatorial binding of transcription factors and architectural proteins, such as CTCF and cohesin[32,33,46]. The binding landscape of CTCF and cohesin was dynamic between cell types, and their density in occupied regions presumably contributed to changes in chromatin interactions and gene expression[48,50–52,58]. Of special note, we infer that Matr3 stabilizes the binding of CTCF and Rad21 to genomic regions. In the absence of Matr3 chromatin undergoes structural changes that promote differentiation.

**Matr3's role in maintaining genome structure provides an alternative perspective for ALS**. Matr3 is a nuclear scaffold protein to which multiple functions have been ascribed. Prior proteomic studies have revealed apparent association with proteins involved in RNA metabolism, nuclear structure, and chromatin. A yeast two-hybrid screen using a fetal brain cDNA library identified 33 unique nuclear proteins that interact with Matr3, the majority of which are involved in RNA processing and chromatin remodeling[19]. Similarly, we found that Matr3 was associated with many chromatin remodeling factors, including CTCF, cohesin, and heterochromatin proteins, as well as RNA binding proteins, in MEL cells.

Attention in the literature has focused on RNA binding properties of Matr3, which have been implicated in RNA processing[21,34]. The potential role of Matr3 in RNA metabolism has received support from studies of amyotrophic lateral sclerosis (ALS), a neurodegenerative disorder with diverse underlying genetics[22,55,59]. Rare ALS associated Matr3 mutations have been reported to alter subcellular localization of the protein and affect interactions with other proteins thought to function in mRNA biogenesis and export[60,61]. Although the significance of these findings is uncertain, the deletion of RNA recognition motifs (RRMs) in Matr3 led to its redistribution into nuclear granules, yet an effect on dose-dependent toxicity of Matr3 in primary neurons was not striking[55]. The lack of a persuasive link between disease causality and RNA-binding properties of Matr3 raises the possibility of other potential mechanisms. In addition, physical association of RNA binding proteins with cohesin has been observed and thus the possibility of a mechanism other than aberrant splicing of the cohesin subunit gene was proposed to explain the loss of sister chromatid cohesion following depletion of RNA processing factors[62]. In this study, we uncovered a distinct chromatin-associated role for Matr3 in regulating 3D genome organization, which suggests that ALS-associated Matr3 mutations may perturb chromatin and gene expression in a more direct manner, as distinguished from the conventional view of its involvement in RNA processing. Additionally, chromatin structure and RNA-processing may be more interconnected than generally appreciated.

**Matr3 is involved in the maintenance of chromatin structure and regulation of gene expression during development**. Due to its spatial proximity to other proteins involved in controlling gene expression, Matr3 has also been implicated in gene regulatory functions, such as DNA replication and transcription[20,23]. Despite its association with these processes, it has remained unclear whether Matr3 influences global gene regulation. Our findings demonstrate that depletion of Matr3 has a broad effect on chromatin organization, and provide unique insights into the impact of a critical inner nuclear protein on the spatial separation of the genome (Fig. 8a).

Proper gene expression and development require the combinatorial activities of transcriptional regulatory factors and coordinated chromatin repositioning. Recent studies have shown that nuclear membrane proteins, such as nuclear lamina, play a critical role in developmental gene expression by regulating spatial positioning of genomic loci[11,16], but much less have been revealed regarding the role of inner nuclear proteins. Chromatin occupancy of CTCF/cohesin changes during differentiation, ultimately affecting dynamic chromatin contacts that regulate gene expression. Our data demonstrate that the nucleoplasmic protein Matr3 stabilizes the binding of the architectural proteins (CTCF and cohesin) to chromatin and serves to maintain chromatin structure (Fig. 8b). We speculate that Matr3 negatively regulates cell fate transitions by maintaining cellular state through fine-tuning the binding of CTCF/cohesin to chromatin and associated 3D interactions. Our work reveals a previously

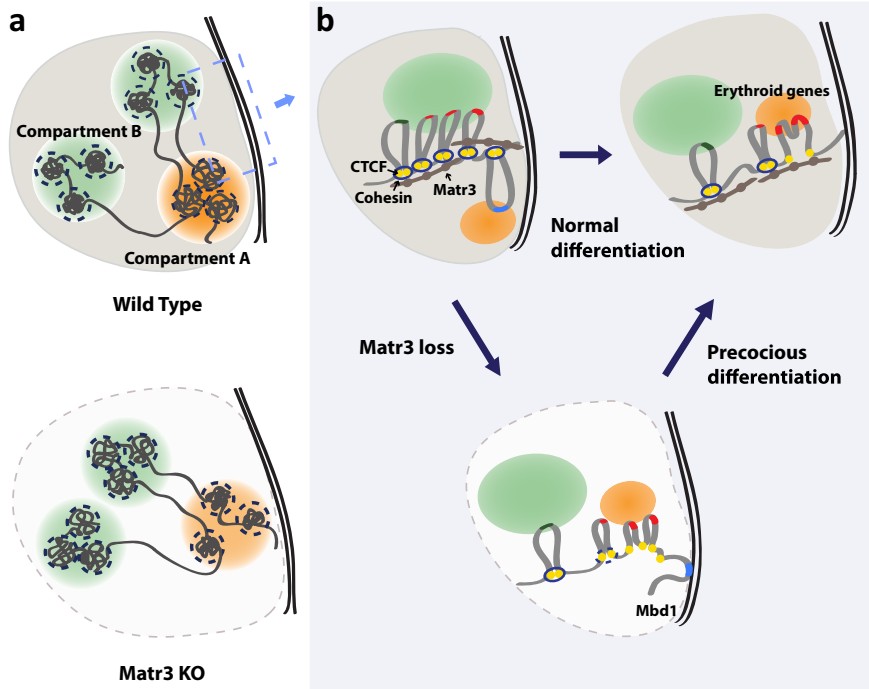

**Fig. 8 Matr3 maintains chromatin structure and coordinates regulation of gene expression and differentiation. a** Matr3 is required to organize chromosomal structure and maintain compartmentalization. The strength of interactions between the B compartments (green) became stronger whereas the contacts between the A-type domains (orange) were reduced in *Matr3* KO cells. In the local chromatin compaction within the TADs indicated by the dashed circles, the intra-TAD contact frequency between compartment B were decreased while the interaction frequency between compartment A increased. **b** The region of the left dashed box is shown on the right. Matr3 interacts with CTCF and cohesin, and Matr3 loss destabilizes their binding to a subset of chromatin regions, which is necessary for proper gene regulation (e.g. Mbd1, marked in blue). The resulting altered chromatin structure resembles that of more differentiated cells and appears to accelerate cell maturation by modulating regulatory regions more accessible. Differentiation-specific erythroid genes are shown in red.

unrecognized role of Matr3 in chromatin organization and responses to developmental cues.

## Methods

**Cell culture**. Mouse erythroleukemia (MEL) cells were cultured in Dulbecco's Modified Eagle's Medium (DMEM) with 10% fetal calf serum, 1% L-glutamine, and 2% penicillin/streptomycin at 37 °C in a humidified atmosphere of 5% $CO_2$[63]. A total of 2% DMSO was used to induce cells differentiation. MEL subclones carrying the BirA enzyme and the tagged version of Matr3 were generated as described[64]. G1ER cells were cultured in Iscove's Modified Dulbecco's Medium (IMDM) with 2% penicillin/streptomycin, $124 \times 10^{-4}$ monothioglycerol, 15% FCS, 2 U/ml recombinant human erythropoietin (EPO), and 50 ng/mL recombinant SCF or SCF-containing media. Cell differentiation was induced by treatment with $10^{-7}$ M β-estradiol to activate estrogen-inducible Gata1/ER fusion protein. Mouse embryonic stem (ES) cells (CJ7) were maintained on mouse embryonic fibroblasts (Gibco MEFs) feeders in standard ES medium (DMEM; Dulbecco's modified Eagle's medium, Thermo Fisher Scientific) supplemented with 15% heat-inactivated fetal calf serum (FCS) (Omega Scientific), 0.1 mM 2-mercaptoethanol (Sigma), 2 mM L-glutamine (Thermo Fisher Scientific), 0.1 mM non-essential amino acid (Thermo Fisher Scientific), 1% of nucleoside mix (Merck Millipore), 50 U/ml Penicillin/Streptomycin (Thermo Fisher Scientific), 1000U/ml recombinant leukemia inhibitory factor(LIF/ESGRO) (Merck Millipore). For all the analysis including RNA-seq and ChIP-seq, ES cells were passed two times on 0.1% gelatin coated plates without feeders. To differentiate mouse ES cells into embryoid bodies (EB), mESCs were passed two times on 0.1% gelatin coated plates without feeders, and a single-cell suspension containing the 50,000 cells/ml to be plated was prepared. Cells were differentiated in Iscove's Modified Dulbecco Media (IMDM) supplemented with 10% heat-inactivated fetal calf serum, 2 mmol L-glutamine, $4.5 \times 10^{-4}$ mol/L monothioglycerol, 0.5 mmol/L ascorbic acid, 200 μg/mL transferrin (Roche), 5% protein free hybridoma media (PFHM-II; Invitrogen) and 50 U/ml Penicillin/Streptomycin. Media was changed at day2 before the EB collection on Day4.

**Immunohistochemistry**. The endogenous expression of Matr3 and HP1α proteins was detected as described[65]. Briefly, cells were fixed using 4% paraformaldehyde in PBS. Cell membranes were permeabilized with 0.5% Triton X-100 in PBS, and nonspecific immunobinding sites were blocked with 4% IgG-free BSA for 1 h at room temperature. Cells were incubated with primary antibodies to Matr3 (Abcam,

ab84422, 1:100), or HP1α (Abcam, ab203432, 1:100). 4′,6-Diamidino-2-pheny-lindole (Sigma) were added as needed.

**Imaging and image analysis**. Immunohistochemistry experiments of Matr3 and HP1α were imaged on a spinning disk confocal microscope (Nikon) and super-resolution structured illumination (SR-SIM) combined with Zeiss LSM710 microscope. Puncta size was measured with Fiji software. Confocal images were cropped and enhanced in Adobe Illustrator and Adobe Photoshop for compilation of figures.

**Histology**. Cytocentrifuge preparations of parental and *Matr3* KO cells at various stages of differentiation were stained with May-Grunwald Giemsa for general morphology. Cell size was measured with Fiji software.

**Fluorescence in situ hybridization (FISH)**. Slides were aged in a 2xSSC solution, dehydrated through a series of alcohols, and air dried. BAC probe RP23-7J22 for *Mbd1* located at 18qE2 was added to the slide and co-denatured at 72 °C for 2 min. They were left to hybridize at 37 °C for 48 h in a humidified chamber. The slides were then washed in 50% formamide, 2xSSC for 10 min at 45 °C. DAPI was applied under a glass coverslip and hybridization signals were viewed on an Olympus AX-70 fluorescent microscope system.

**RNA Isolation and qRT-PCR**. RNA was extracted using TRIzol reagent (Thermo Fisher) and the RNeasy Plus Mini Kit (Qiagen). RNA was reverse-transcribed using iScript cDNA Synthesis Kit (Bio-Rad) and quantitative PCR was performed using the iQ SYBR Green Supermix (Bio-Rad) and CFX Real-Time PCR Detection System (Bio-Rad). Following primer sequences were used for qRT-PCR: *Hbb-b1*, TTTAACGATGGCCTGAATCACTT and CAGCACAATCACGATCATATTGC; *Slc4a1*, ATGGCCTCAAAGTGTCCAAC and TCAGCGTGGTGATCTGAGAC; *Mbd1*, AACTGAGCTCTCCCTTAAAGG and TGACTGCTGTCCACTCCTCTG; *Gapdh*, AAATTCAACGGCACAGTCAAG and CACCCCATTTGATGTTAGTGG.

**Immunoprecipitation and Western Blotting**. Nuclei were isolated from MEL cells and lysed to make nuclear extracts. Nuclear extracts were then incubated with the indicated antibodies overnight at 4 °C. Protein G/A magnetic beads (Thermo Scientific) equilibrated with IP buffer (20 mM HEPES pH 7.9, 25% glycerol,

200 mM NaCl, 1.5 mM MgCl$_2$, 0.2 mM EDTA, 0.02 % NP-40) were added and the mixture was incubated for an additional 2 h at 4 °C. Beads were washed four times for 15 min and eluted by boiling in 2X Laemmli sample buffer.

The protein expression was detected as described[66]. Briefly, cells were lysed in RIPA buffer (Boston Bioproducts) containing 1 mM DTT, 1 mM PMSF, and protease inhibitors and analyzed by SDS-PAGE and Western blotting using specific antibodies. For immunoprecipitation assays, protein extracts were mixed with Laemmli buffer and boiled at 95 °C for 5 min, followed by SDS-PAGE. Following antibodies diluted 1:1000 were used: Esco2 (bethyl, A301-689A), Matr3 (Abcam, ab84422), CTCF (Abcam, ab70303), Rad21 (Abcam, ab992), Smc3 (Abcam, ab9263), Mbd1 (Abcam, ab187734). Uncropped blots are provided in Source data.

**Hi-C library preparation**. In total 5 million MEL cells were crosslinked with 1% formaldehyde for 10 min and then quenched with glycine. Cells were lysed and then digested with DpnII overnight at 37 °C. Sticky ends were filled with dNTPs containing biotin-14-dATPs at 23 °C for 4 h. Furthermore, blunt ends were ligated using T4 DNA ligase at 16 °C for 4 h. Ligation products were treated with proteinase K at 65 °C overnight to reverse cross-linking and then purified using phenol-chloroform extraction. Ligation products were confirmed by agarose gel. Biotins were removed from un-ligated ends and then fragmented to average size of 200 bp by sonication. Fragmented DNAs were size-selected up to 350 bp using AMPure XP beads (Beckman Coulter, catalog no. A63881). Fragments were end-repaired, A-tailed and then biotin-tagged ends were pulled down using streptavidin beads. Illumina TruSeq adapted were ligated, cleaned-up and amplified using Illumina PCR master mix. Final Hi-C library products were sequenced using PE50 on HiSeq 2500 or NextSeq500.

**Hi-C data processing**. Hi-C PE50 fastq files were mapped to mm10 mouse reference genome using distiller-nf mapping pipeline (https://github.com/mirnylab/distiller-nf) and then downstream analysis were done by 'pairtools' (https://github.com/mirnylab/pairtools) and 'cooltools' (https://github.com/mirnylab/cooltools). Briefly, fastq reads were mapped using bwa-mem, and then classified, de-duplicated and low-quality reads were filtered out using pairtools (phred score<30) to achieve valid pairs. Valid pairs were binned at 1 kb, 2 kb, 5 kb, 10 kb, 25 kb, 50 kb, 100 kb, 250 kb, 500 kb, and 1 Mb resolutions to generate interaction matrices for further analysis. After confirming the consistent results from two biological replicates, the results of combined data are shown.

*Compartment analysis*. A and B compartments were defined by eigen vector decomposition on 100 kb binned Hi-C data using the call-compartments function of the cooltools. Calculated PC1 values were used for A and B compartments: positive values for A and negative values for B compartments.

To measure compartmentalization strength, observed/expected interaction frequencies were calculated on 100 kb binned Hi-C matrices using compute-expected function of cooltools (https://github.com/mirnylab/cooltools) to correct for average distance decay of each dataset. Then, ordered matrices for each chromosome within a dataset were aggregated. The aggregated matrix divided in 50 bins and plotted as a saddle plot using compute-saddle --strength function of the cooltools. Strength of A compartmentalization (lower right corner) was defined as the ratio of (A–A/A–B) interactions. Strength of B compartmentalization (upper left corner) was defined as the ratio of (B–B/A–B) interactions. The numbers of the heatmaps indicate the average compartment strength quantified by calculating the ratio of homotypic (A-A or B-B) to heterotypic (A-B) compartment interactions of the top 20% sorted EV1 values. The ratio values were calculated by averaging of 10 bins in each corner of the saddle plot. Chromosomes X, Y, and M are excluded from saddle plot analysis.

*Quantification of changes in compartmentalization strength*. Hi-C reads were aligned to the mm10 genome (GRCm38) with HiC-Pro (version 2.11.1). Low-quality read pairs with MAPQ lower than 30 were discarded, and only one unique read pair was kept. The resulted "allValidPairs" files were used as input to generate the.mcool files using the cooler package (version 0.8.6)[67], and hi-glass[68] was used for the visualization. Specifically, the '.allValidPairs' files from HiC-Pro were used to obtain the raw contact matrices at resolution (i.e bin size) of 5 kb with the 'cload pairs' function. The raw contact matrices were then normalized using the 'balance' function. The normalized matrices were zoomed to other resolutions in '.mcool' format by the 'zoomify' function. The obs/exp contact matrices were generated by juicer tools (version 1.7.5)[69] at 50 kb resolution. Data on chrY were excluded. Then, A/B compartments were identified using the "runHiCpca.pl" script in HOMER[70] at the resolution of 50 kb. The "±" sign of the compartment was determined by TSS enrichments, where compartments showing enrichment of TSS were assigned the '+' sign and those showing depletion of TSS were assigned the '-' sign. To visualize the changes in compartmentalization strength, we generated and compared the saddle plots. First, for each chromosome, we removed the 1% genomic bins with the lowest sequencing coverage in order to remove the bias caused by insufficient coverage. Then we ranked the remaining genomic bins by the PC1 scores from high to low. We reordered the rows and columns of the contact matrix according to the same ordering. Then the contact map was coarse-grained into a 100*100 matrix, where the element (m, n) represents the mean interaction frequency

between bins of the m-th percentile and the n-th percentile. The average of the coarse-grained contact matrices from all chromosomes were then plotted as the saddle plot. The obs/exp contact matrices at 50 kb were used for this analysis. To visualize the differences in saddle plots of two samples, we plotted the fold change of the two contact matrices in an element-wise way after log2 transformation. To quantify the compartmentalization strength changes, we used a metric named compartment score, which was defined in[71] and used to quantify the degree of preference that a genomic bin interacts with regions of the same compartment type. Specifically, compartment score were calculated for 50 kb genomic bins and defined as the average interactions with other bins of the same compartment type with respect to the average interactions with all other bins. For genomic bins showing no compartmentalization, the compartment score is 1. For bins demonstrating compartmentalization, the score is higher than 1.

*Insulation and TAD boundary pile-up analysis*. Insulation scores for domain boundaries were called using cooltools diamond-insulation function on 25 kb binned Hi-C data using 250 kb window size. One dimension insulation scores pile-up was plotted using 250 kb up- and down-stream regions of insulation scores with a boundary strength >0.1. To observe differences between conditions, 2D interaction heatmaps were plotted using insulation scores. Normalized interaction frequencies (observed/expected) of the regions where insulation scores have a boundary strength > 0.1, were aggregated and TAD boundary plots were generated. The differences between conditions were calculated by taking log2 ratios of the interaction heatmap values. Similarly, interaction heatmap pile-ups for CTCF and Rad21 peaks were plotted by aggregating the observed/expected Hi-C matrices of 250 kb surrounding region of the peaks.

*Identification of TAD numbers*. To reveal the dynamics in the number of TADs, we called TADs from the Hi-C datasets using two methods: Direction Index (DI) and TADbit[30,31]. For both methods, the ICE-normalized Hi-C matrices at 40 kb resolution from HiC-Pro were used as the input. To calculate the DI, we used the HiCtool package (https://github.com/Zhong-Lab-UCSD/HiCtool). TADs were identified from DI by a HMM method in HiCtool.

*Contact frequency within TADs*. TAD boundaries were identified using[29]. To create a set of non-redundant TAD boundaries, we adopted an approach as described in[72]. First, we created a file with the TAD boundaries from all the samples and sorted them by their insulation scores in descending order. Then, we picked one TAD boundary from the top of the list and removed any remaining boundary within 3 bins (3 × 40 kb = 120 kb) of the top TAD boundary. Then, we picked the next TAD boundary on the list and repeated the same process until the entire list was traversed. In total, we identified 3952 unique TAD boundaries. Consensus TADs are generated from the non-redundant TAD boundaries in following steps. First, every pair of adjacent TAD boundaries constructs a potential TAD. Second, there are regions between the TAD boundaries that should not be identified as TADs (e.g regions between two real TADs). To filter these regions out, we overlapped each potential TAD with the TADs identified by the DI method from the samples and those with an overlap ratio higher than 50% were kept. Third, TADs shorter than 6 bins were removed. At the end, there are 3338 consensus TADs used for following analyses.

To show the compartment-specific changes of contact frequency within the TAD, we first classified TADs into A and B compartments by the mean PC1 values of genomic bins within a TAD. If the mean PC1 values is positive then the TAD is classified as locating within the A compartment. The mean contact frequency between the bins within a TAD was calculated in the obs/exp contact map. To identify the TADs that show significant changes in interaction frequency, we calculated the interactions for each condition in the two replicates and performed a paired two-sided t-test. The TADs with P-value more significant than 0.05 and log2(fold change) higher/lower than ± 0.15 would be deemed as significant. Hyper-geometrical distribution was used to compare TADs with significantly altered interactions. The expected value was the expected number of overlapping TADs at a random setting.

*P(s) and derivative plots*. Intra-chromosomal interaction (cis) valid read pairs were used to calculate the contact frequency (P) as a function of genomic distance (s) using cooltools. Derivative plots were calculated and plotted according to each P(s) plot.

*Loop calling and differential loop analysis*. Based on the valid pairs which specify the number of reads for each interaction anchor pair, we binned the anchor coordinates to 8-fragment resolution and summarized the reads at this resolution. Reads were normalized to a distance-normalized interaction frequency (IFnorm)[73,74] according to:

$$\text{IFnorm}(i,j) = \frac{\text{IF}(i,j)}{\text{mean}\{\text{IF}(a,b): \left\lceil \frac{\text{pos}(b)-\text{pos}(a)}{10{,}000} \right\rceil = \left\lceil \frac{\text{pos}(j)-\text{pos}(i)}{10{,}000} \right\rceil\}}$$ where pos(a) is the genomic

position of restriction fragment (RF) a. We next used the HiCCUPS algorithm to call interaction loops. HiCCUPS accepts two parameters p (peak size) and w(window size). It compares the peak region P(i,j) with several types of flanking regions including the "donut" around peak, the top and bottom flanks, and the left and right flanks. The maximum of the several flanks (maxFlank(i,j)) was used to compute a score(i,j):

score(i, j) = P(i, j)/maxFlank(i, j)

Default values of $p = 50$ kb, and $w = 100$ kb were used. Original implementation of HiCCUPS was described in[73]. We used the fast C++ implementation described in HIFI[74] to speed up computation, here setting HIFI to use the fixed resolution binning approach: -method=fixed and -fragmentPerBin=1. Significant loops were generated by comparing score(i,j) to those of distance-controlled randomly permuted IF matrix.

Next to compute differential loops between two samples (s1, s2), we first define a set of "common" regions consisting of loops unique to s1, unique to s2, and shared by s1 and s2 where the the loops' genomic locations intersect. The difference of score(i,j) between s1 and s2 is calculated per common region (i,j) as:

$$\text{diff}(\text{score}(i, j)) = \text{score\_s1}(i, j) - \text{score\_s2}(i, j),$$

where score_s1(i,j) is the score(i,j) in s1 if the (i,j) is a loop in the sample s1, or 1.0 if (i,j) is not a loop in s1. A z-score is computed from the distribution of diff(score(i,j)). Differential loops are identified as $z > 1.5$.

*Visualization of interaction matrix*. Visualization of the IF-norm interaction matrix was provided by HIFI package[74] available at https://github.com/BlanchetteLab/HIFI. We used the function plotHIFIoutput.py with vmin = 0 and vmax = 1.5.

**Affinity purification of protein complexes and mass spectrometry**. Matr3-interacting protein complexes were purified and characterized as described[64]. Parental and *Matr3* KO MEL cells expressing biotinylated Matr3 and BirA were established. Parental and *Matr3* KO cells expressing only BirA were used as controls. Co-eluted proteins were separated by SDS-PAGE and cut out from the gel.

Excised gel bands were cut into approximately 1 mm³ pieces. Gel pieces were then subjected to a modified in-gel trypsin digestion procedure[75]. Sample pieces were washed and dehydrated with acetonitrile for 10 min, followed by removal of acetonitrile. Pieces were then completely dried in a speed-vac. Rehydration of the gel pieces was with 50 mM ammonium bicarbonate solution containing 12.5 ng/μl modified sequencing-grade trypsin (Promega, Madison, WI) at 4 °C. After 45 min, the excess trypsin solution was removed and replaced with 50 mM ammonium bicarbonate solution to just cover the gel pieces. Samples were then placed in a 37 °C room overnight. Peptides were later extracted by removing the ammonium bicarbonate solution, followed by one wash with a solution containing 50% acetonitrile and 1% formic acid. The extracts were then dried in a SpeedVac and stored at 4 °C. Samples were reconstituted in 5 - 10 μl of HPLC solvent A (2.5% acetonitrile, 0.1% formic acid). A nano-scale reverse-phase HPLC capillary column was created by packing 2.6 μm C18 spherical silica beads into a fused silica capillary (100 μm inner diameter x ~30 cm length) with a flame-drawn tip[76]. After equilibrating the column each sample was loaded via a Famos auto sampler (LC Packings, San Francisco CA) onto the column. A gradient was formed and peptides were eluted with increasing concentrations of solvent B (97.5% acetonitrile, 0.1% formic acid). As peptides eluted they were subjected to electrospray ionization and then entered into an LTQ Orbitrap Velos Pro ion-trap mass spectrometer (Thermo Fisher Scientific, Waltham, MA). Peptides were detected, isolated, and fragmented to produce a tandem mass spectrum of specific fragment ions for each peptide. Peptide sequences (and hence protein identity) were determined by matching protein databases with the acquired fragmentation pattern by the software program, SEQUEST (Thermo Fisher Scientific, Waltham, MA)[77]. All databases include a reversed version of all the sequences and the data was filtered to between a one and two percent peptide false discovery rate. The mass spectrometry proteomics data have been deposited to the ProteomeXchange Consortium via the PRIDE[78] partner repository with the dataset identifier PXD028867 and 10.6019/PXD028867.

**Immunoprecipitation of cohesin complex followed by mass spectrometry**

*SMC1A_iTRAQ (SMC1A IP vs IgG IP in WT cells)*. A total of 20 million cells were used to generate 2 mg input protein and immunoprecipitation was performed using 25ug SMC1A antibody or 25ug control IgG as described above. The beads from immunoprecipitation were washed once with IP lysis buffer and twice with IP wash buffer. The beads were resuspended in 20 μL of wash buffer, followed by 90 μL digestion buffer (2 M urea, 50 mM Tris HCl) and then 2 μg sequencing grade trypsin was added, followed by 1 hour of shaking at 700 rpm. The supernatant was removed and placed in a fresh tube. The beads were then washed twice with 50 μl digestion buffer and combined with the supernatant. The combined supernatants were reduced (2 μl 500 mM dithiothreitol, 30 minutes, room temperature) and alkylated (4 μl 500 mM iodoacetamide, 45 min, dark), and a longer overnight digestion was performed: 2 μg (4 μl) trypsin, shaken overnight. The samples were then quenched with 20 μl 10% formic acid and desalted on 10 mg Oasis cartridges. Desalted peptides were labeled with iTRAQ reagents according to the manufacturer's instructions (AB Sciex, Foster City, CA).

Peptides were dissolved in 30 μl 0.5 M TEAB pH 8.5 solution (Sigma-Aldrich) and labeling reagent was added in 70 μl of ethanol. After a 1-h incubation, the reaction was stopped with 50 mM Tris-HCl pH 7.5. Differentially labeled peptides were mixed and subsequently desalted on a 10 mg SepPak column. 50% of the sample was used for SCX fractionation as described in[79], with 6 pH steps (buffers- all contain 25% acetonitrile) as follows: 1, ammonium acetate 50 mM pH 4.5; 2, ammonium acetate 50 mM pH 5.5; 3, ammonium acetate 50 mM pH 6.5; 4, ammonium bicarbonate 50 mM pH 8; 5, ammonium hydroxide 0.1% pH 9; 6, ammonium hydroxide 0.1% pH 11. Empire SCX disk used to make StageTips as described[79].

Reconstituted peptides from each fraction were separated on an online nanoflow EASY-nLC 1000 UHPLC system (Thermo Fisher Scientific) and analyzed on a benchtop Orbitrap Q Exactive Plus mass spectrometer (Thermo Fisher Scientific). The peptide samples were injected onto a capillary column (Picofrit with 10 μm tip opening/75 μm diameter, New Objective, PF360-75-10-N-5) packed in-house with 20 cm C18 silica material (1.9 μm ReproSil-Pur C18-AQ medium, Dr. Maisch GmbH) and heated to 50 °C in column heater sleeves (Phoenix-ST) to reduce backpressure during UHPLC separation. Injected peptides were separated at a flow rate of 200 nl min⁻¹ with a linear 120 min gradient from 100% solvent A (3% acetonitrile, 0.1% formic acid) to 30% solvent B (90% acetonitrile, 0.1% formic acid), followed by a linear 9 min gradient from 30% solvent B to 60% solvent B and a 1 min ramp to 90% B. The Q Exactive instrument was operated in the data-dependent mode acquiring higher-energy collisional dissociation (HCD) tandem mass spectrometry (MS/MS) scans (R = 17,500) after each MS1 scan (R = 70,000) on the 12 top most abundant ions using an MS1 ion target of $3 \times 10^6$ ions and an MS2 target of $5 \times 10^4$ ions. The maximum ion time utilized for the MS/MS scans was 120 ms; the HCD-normalized collision energy was set to 27; the dynamic exclusion time was set to 20 s; and the peptide match and isotope exclusion functions were enabled.

All mass spectra were processed using the Spectrum Mill software package v6.0 prerelease (Agilent Technologies), which includes modules developed by us for iTRAQ-based quantification. For peptide identification MS/MS spectra were searched against the human Uniprot database (UniProt.human.20141017.RNFISnr.150contams) to which a set of common laboratory contaminant proteins was appended. Search parameters included ESI-QEXACTIVE-HCD scoring parameters, trypsin enzyme specificity with a maximum of two missed cleavages, 40% minimum matched peak intensity, ± 20 ppm precursor mass tolerance, ± 20 ppm product mass tolerance, and carbamidomethylation of cysteines and iTRAQ labeling of lysines and peptide N termini as fixed modifications. Allowed variable modifications were oxidation of methionine, N-terminal acetylation, pyroglutamic acid (N-termQ), deamidated (N), pyro carbamidomethyl Cys (N-termC), with a precursor MH + shift range of −18–64 Da. Identities interpreted for individual spectra were automatically designated as valid by optimizing score and delta rank1-rank2 score thresholds separately for each precursor charge state in each liquid chromatography-MS/MS while allowing a maximum target-decoy-based false-discovery rate (FDR) of 1.0% at the spectrum level.

In calculating scores at the protein level and reporting the identified proteins, redundancy is addressed in the following manner: the protein score is the sum of the scores of distinct peptides. A distinct peptide is the single highest scoring instance of a peptide detected through an MS/MS spectrum. MS/MS spectra for a particular peptide may have been recorded multiple times, (i.e. as different precursor charge states, isolated from adjacent SCX fractions, modified by oxidation of Met) but are still counted as a single distinct peptide. When a peptide sequence >8 residues long is contained in multiple protein entries in the sequence database, the proteins are grouped together and the highest scoring one and its accession number are reported. In some cases when the protein sequences are grouped in this manner there are distinct peptides which uniquely represent a lower scoring member of the group (isoforms or family members). Each of these instances spawns a subgroup and multiple subgroups are reported and counted towards the total number of proteins. iTRAQ ratios were obtained from the protein-comparisons export table in Spectrum Mill. To obtain iTRAQ protein ratios the median was calculated over all distinct peptides assigned to a protein subgroup in each replicate. The data have been published in[80].

*SMC1A_TMT6 (SMC1A IP in WT vs STAG2 KO cells)*. A total of 18 million cells were used to generate 4 mg input protein and immunoprecipitation was performed using 31.5 μg SMC1A antibody as described above. The beads from immunoprecipitation were washed once with IP lysis buffer, twice with IP wash buffer, then once with PBS. The beads were resuspended in 20 μL of PBS, followed by 90 μL digestion buffer (2 M urea, 50 mM Tris HCl) and then 2 μg sequencing grade trypsin was added, followed by 1 h of shaking at 700 rpm. The supernatant was removed and placed in a fresh tube. The beads were then washed twice with 50 μl digestion buffer and combined with the supernatant. The combined supernatants were reduced (2 μl 500 mM dithiothreitol, 30 min, room temperature) and alkylated (4 μl 500 mM iodoacetamide, 45 min, dark), and a longer overnight digestion was performed: 2 μg (4 μl) trypsin, shaken overnight. The samples were then quenched with 20 μl 10% formic acid and desalted on 10 mg Oasis cartridges.

Desalted peptides were labeled with TMT6 reagents Lot# RA230200 (Thermo Fisher Scientific) Peptides were dissolved in 25 μl fresh 100 mM HEPES buffer. The labeling reagent was resuspended in 42 μl acetonitrile and 10 μl added to each sample as described below. After 1 hour incubation the reaction was stopped with 8 μl 5 mM hydroxylamine.

In total 50% of the combined sample was used for basic reversed phase fractionation as described in ref. [79] with 6 cuts as follows: 1, 10% ACN; 2, 15% ACN; 3, 20% ACN; 4, 35% ACN; 5, 50% ACN; 6, 80% ACN. The fractions were then concatenated int 3 combining fractions (1 + 4), (2 + 5) and (3 + 6) to create three fractions. Empore SDB disk used to make StageTips as described[79].

Reconstituted peptides from each fraction were separated on an online nanoflow EASY-nLC 1000 UHPLC system (Thermo Fisher Scientific) and analyzed on a benchtop Orbitrap Q Exactive Plus mass spectrometer (Thermo Fisher Scientific). The peptide samples were injected onto a capillary column (Picofrit with 10 μm tip opening/75 μm diameter, New Objective, PF360-75-10-N-5) packed in-house with 20 cm C18 silica material (1.9 μm ReproSil-Pur C18-AQ medium, Dr. Maisch GmbH) and

heated to 50 °C in column heater sleeves (Phoenix-ST) to reduce backpressure during UHPLC separation. Injected peptides were separated at a flow rate of 200 nl min$^{-1}$ with a linear 120 min gradient from 100% solvent A (3% acetonitrile, 0.1% formic acid) to 30% solvent B (90% acetonitrile, 0.1% formic acid), followed by a linear 9 min gradient from 30% solvent B to 60% solvent B and a 1 min ramp to 90% B. The Q Exactive instrument was operated in the data-dependent mode acquiring higher-energy collisional dissociation (HCD) tandem mass spectrometry (MS/MS) scans (R = 17,500) after each MS1 scan (R = 70,000) on the 12 top most abundant ions using an MS1 ion target of $3 \times 10^6$ ions and an MS2 target of $5 \times 10^4$ ions. The maximum ion time utilized for the MS/MS scans was 120 ms; the HCD-normalized collision energy was set to 29; the dynamic exclusion time was set to 20 s; and the peptide match and isotope exclusion functions were enabled.

All mass spectra were processed using the Spectrum Mill software package v6.0 prerelease (Agilent Technologies), which includes modules developed by us for TMT-based quantification. For peptide identification MS/MS spectra were searched against the human Uniprot database (UniProt.human.20141017.RNFISnr_CanCom.150contams) to which a set of common laboratory contaminant proteins was appended. Search parameters included ESI-QEXACTIVE-HCD scoring parameters, trypsin enzyme specificity with a maximum of two missed cleavages, 40% minimum matched peak intensity, ± 20 ppm precursor mass tolerance, ± 20 ppm product mass tolerance, and carbamidomethylation of cysteines and TMT6 labeling of lysines and peptide N termini as fixed modifications. Allowed variable modifications were oxidation of methionine, N-terminal acetylation, pyroglutamic acid (N-termQ), deamidated (N), pyro carbamidomethyl Cys (N-termC), with a precursor MH + shift range of −18–64 Da. Identities interpreted for individual spectra were automatically designated as valid by optimizing score and delta rank1-rank2 score thresholds separately for each precursor charge state in each liquid chromatography-MS/MS while allowing a maximum target-decoy-based false-discovery rate (FDR) of 1.0% at the spectrum level. The data have been published in[80].

**Genomic deletion by CRISPR/Cas9.** The clustered regularly interspaced short palindromic repeats (CRISPR)/CRISPR-associated (Cas) 9 nuclease system was used to generate deletion mutations in MEL, G1ER, and mESC cells as described[81]. Briefly, paired single guide RNAs (sgRNAs) for site-specific cleavage of genomic regions were designed according to the described guidelines, and were selected to minimize off-target effects based on publicly available online tools (http://crispr.mit.edu/). 10 μM guide oligo and 10 μM reverse complement oligo were mixed with 1 × T4 ligation buffer and 5U of T4 Polynucleotide Kinase (PNK), and annealed in a thermo cycler using the following parameters: 37 °C for 30 min, 95 °C for 5 min, and ramp down to 25 °C at 5 °C/min. Annealed oligos were subsequently ligated into pX330 vector using a Golden Gate assembly strategy. Ligation reaction mixture was prepared as follows: 100 ng vector, 1 μl annealed oligos (1 μM), 20 U of BbsI restriction enzyme, 1 mM ATP, 5 μg BSA, 750 U of T4 DNA ligase, 1 × 750 U of T4 DNA ligase, and H$_2$O to final volume of 50 μl. Samples were then incubated in a thermo cycler using the following parameters: 20 cycles of 37 °C for 5 min and 20 °C for 5 min, followed by 80 °C for 20 min. A pair of CRISPR/Cas9 constructs (5 μg each) targeting each flanking region of the deletion site were transfected into $2 \times 10^6$ cells with 0.5 μg of GFP expression plasmid using an electroporation system (Harvard Apparatus). The top 1-5% of GFP positive cells were sorted by FACS 24–48 hours post-transfection. Single cell derived clones were isolated and screened for biallelic deletion of target genomic sequences. PCR amplicons were subcloned into a plasmid vector and subjected to Sanger sequencing to confirm deletion. Following sgRNAs were used:

| Target | Name | Sequences |
|---|---|---|
| *Matr3* KO | Matr3-KO-L1-F | CACCGAGGCACGTGACGTACGCGGC |
| | Matr3-KO-L1-R | AAACGCCGCGTACGTCACGTGCCTC |
| | Matr3-KO-L2-F | CACCGCGTCACGTGCCTACCCCGCG |
| | Matr3-KO-L2-R | AAACCGCGGGGTAGGCACGTGACGC |
| | Matr3-KO-R1-F | CACCGTATCGAGGTGATGGTCGTAT |
| | Matr3-KO-R1-R | AAACATACGACCATCACCTCGATAC |
| | Matr3-KO-R2-F | CACCGATCGGGTTTTATCGAGGTGA |
| | Matr3-KO-R2-R | AAACTCACCTCGATAAAACCCGATC |
| *Mbd1* KO | Mbd1-KO-L1-F | CACCGCTCGGCCCACTCCGAATTT |
| | Mbd1-KO-L1-R | AAACAAATTCGGAGTGGGCCGAGC |
| | Mbd1-KO-L2-F | CACCGTGGAGACCGAAATTCGGAGT |
| | Mbd1-KO-L2-R | AAACACTCCGAATTTCGGTCTCCAC |
| | Mbd1-KO-R1-F | CACCGGTTGACAAAATTCTCGTAC |
| | Mbd1-KO-R1-R | AAACGTACGAGAATTTTGTCAACC |
| | Mbd1-KO-R2-F | CACCGCTGTGGGAAAACGGGGTCGT |
| | Mbd1-KO-R2-R | AAACACGACCCCGTTTTCCCACAGC |
| *Mbd1* Prox deletion | Mbd1-prox-L1-F | CACCGCGGGTACCAATCCTGAAGA |
| | Mbd1-prox-L1-R | AAACTCTTCAGGATTGGTACCCGC |
| | Mbd1-prox-R1-F | CACCGGTTCCTAGTCGGAGCCCCAA |
| | Mbd1-prox-R1-R | AAACTTGGGGCTCCGACTAGGACC |

**ATAC-seq.** In total 50,000 cells were washed and lysed using cold lysis buffer (10 mM Tris-HCl, pH 7.4, 10 mM NaCl, 3 mM MgCl2 and 0.1% IGEPAL CA-630). The pellet was then resuspended in the transposition reaction mix (25 μL 2× TD buffer, 2.5 μL Tn5 Transposes (Illumina) and 22.5 μL nuclease-free water) and purified using a Qiagen MinElute Kit. Transposed DNA fragments were amplified by PCR and libraries were sequenced on Illumina HiSeq 2500 with paired-end reads.

Sequencing reads were trimmed for adapter sequences and low quality bases using Cutadapt[82], and then aligned to the assembly of the mouse genome mm10 using Bowtie2[83] with the parameter –X 2000 allowing fragment length up to 2 kb to aligned[84]. Reads were filtered for properly paired reads, and duplicates and reads mapped to the mitochondria were removed. All reads aligned to the + strand were offset by + 4 bp and reads aligned to the – strand were offset by –5 bp[85]. Peaks were called subsequently using MACS2[86] with following parameters (--nomodel --shift 37 --extsize 73). For peak comparison, differential ATAC-seq peaks were obtained using MAnorm[38] and the results of two biological replicates were combined to generate a more stringent peak list. Known motifs enriched in altered peak regions were identified using HOMER[87]. Proximal regions of transcription start sites (TSS) are ±2 kb windows centered by RefSeq TSS locations[88], and the remaining regions were considered as distal sites.

**RNA-seq.** Total RNA was isolated from MEL cells using TRIzol reagent (Thermo Fisher) and the RNeasy Plus Mini Kit (Qiagen), and ribosomal RNAs were depleted to prepare the RNA-seq library using NEBNext Ultra Directional RNA Library Prep Kit for Illumina. Paired-end 100 bp reads were generated using Illumina HiSeq 2500 sequencing platform. The sequencing reads were mapped to the mm10 mouse reference genome using STAR (v2.5.4a)[89], raw counts were produced using HTseq[90], and differential expression analyses were performed using edgeR[91] and DESeq2[92]. edgeR was also used to calculate the normalized counts and z-scores, and DESeq2 was used for the fragments per kilobase per million reads mapped (FPKM) values.

Differential alternative splicing isoforms were identified using Mixture of Isoforms (MISO) framework[93]. Splicing event categories, such as skipped exons (SE), alternative 3'/5' splice sites (A3SS, A5SS), mutually exclusive exons (MXE), and retained introns (RI), were downloaded from http://genes.mit.edu/burgelab/miso/ as mouse genome (mm10) v2, and the outputs were combined and compared to gene expression.

Total RNA was isolated from ES cells using RNeasy Plus Mini Kit (Qiagen). RNA sequencing libraries were prepared using Roche Kapa mRNA HyperPrep sample preparation kits from 100 ng of purified total RNA according to the manufacturer's protocol. The finished dsDNA libraries were quantified by Qubit fluorometer, Agilent TapeStation 2200, and RT-qPCR using the Kapa Biosystems library quantification kit according to manufacturer's protocols. Uniquely indexed libraries were pooled in equimolar ratios and sequenced on Illumina NextSeq500 with single-end 75 bp reads.

Knockout and wildtype samples in mouse MEL and ES cell are processed with default settings using HISAT2[94] which includes alignment, marking of duplicates, and quantification of genes according to the mm10 genome assembly. The gene expression in raw counts was produced using a Python script downloaded from https://ccb.jhu.edu/software/stringtie/dl/prepDE.py to prepare for differential expression analysis. Next, the counts table was used as input for DESeq2[92] analysis to find differential expressed genes between knockout and wildtype in each day. Principle component analysis (PCA) was performed to verify that the samples are correctly clustered according to the expression profile.

*Gene ontology analysis.* Statistically significant differential expressed genes (P adjusted < 0.05) were selected for gene ontology (GO) term enrichment analysis[95]. Where the number of differentially expressed (DE) genes was less than 100 genes, we performed a coexpression expansion of the DE genes using SEEK[96] (http://seek.princeton.edu/) to expand the list to an additional 100 coexpressed genes. The total was then used for gene enrichment analysis. To compute enrichment, we performed a hypergeometric distribution test:

$$\Pr(X \geq k) = \sum_{i=k \dots K} \frac{\binom{K}{i}\binom{N-K}{n-i}}{\binom{N}{n}}$$

Where N is the total number of genes, K is the number of genes in each GO term, k is the number of overlapped genes between a DE gene list and a GO term, and n is the number of genes in the DE list. Such a test was performed between the DE list and each of GO biological process terms with a minimum term size of 10 genes, and using only experimentally derived gene annotations in GO[97]. Multiple hypothesis testing correction was performed using the qvalue R package[98].

**ChIP-seq.** Cells were crosslinked and nuclei were prepared using truChIP Chromatin Shearing Reagent Kit (Covaris). Chromatin was sonicated to around 200–500 bp in shearing buffer (10 mM Tris-HCl pH 7.6, 1 mM EDTA, 0.1% SDS) using a Covaris E220 sonicator. Sheared chromatin was diluted in RIPA buffer (10 mM Tris-HCl pH 7.4, 150 mM NaCl, 1 mM EDTA, 0.1% SDS, 1% Triton X-100, 0.1% sodium deoxycholate and protease inhibitor), and incubated with anti-body at 4 °C overnight. Protein A/G Dynabeads (Thermo Fisher Scientific) were added to the ChIP reaction and incubated for 3 h at 4 °C. After incubation, beads

were washed twice with RIPA buffer, twice with RIPA buffer with 0.3 M NaCl, twice with LiCl wash buffer (10 mM Tris-HCl, 1 mM EDTA, 250 mM LiCl, 0.5% sodium deoxycholate, 0.5% NP-40, pH 8.0), and twice with TE buffer (10 mM Tris-HCl pH 8.0, 1 mM EDTA, pH 8.0). The chromatin was eluted in elution buffer (1% SDS, 10 mM EDTA, 50 mM Tris-HCl, pH 8.0) and reverse-crosslinked at 65 °C overnight. ChIP DNA was treated with RNaseA and protease K, and purified using phenol-chloroform extraction and QIAquick spin columns (Qiagen). Following antibodies were used: CTCF (Abcam, ab70303), Rad21 (Abcam, ab992), H3K4me1 (ab8895, Abcam), H3K27ac (ab4729, Abcam), H3K36me3 (ab9050, Abcam). 1 to 5 µg antibody was used per ChIP reaction.

Purified DNA was processed to generate libraries using the NEBNext ChIP-seq Library Prep Master Mix, NEBNext Ultra DNA Library Prep Kit for Illumina (New England BioLabs), ThruPLEX DNA-Seq Kit (Rubicon Genomics), or Accel-NGS 2 S Plus DNA Library Kit (Swift Biosciences) according to the manufacturer's instructions. Prepared libraries were validated by Bio-analyzer and ChIP sequencing was performed using Illumina HiSeq2500 or NextSeq500. Sequencing reads were aligned to the assembly of the mouse genome mm10 using Bowtie2[83] with the default parameters, and duplicate reads were removed using Picard Tools (http://broadinstitute.github.io/picard/). Peaks were called using Model-based Analysis of ChIP-Seq (MACS2)[86] and significant enrichment regions were determined by q-values (0.01).

Differential peaks were obtained using MAnorm, which quantitatively compares ChIP-seq datasets[38] and the results of at least two independent experiments were combined to generate a more stringent peak list. After classifying peak regions, enrichment of the peaks in each experiment was measured using two independent methods, MACS2 output[86] and deepTools[99], and the results were confirmed in 2-3 biological replicates. Differentially bound regions were identified using Diffbind[39] using at least two biological replicates. Association between a set of genomic regions and the transcriptional activity of target genes was assessed by performing Fisher's exact test[100].

*Distance from TSS to enhancers.* To see how the enhancer landscape changes with the gene expression landscape, we compared the distance between TSSs of significantly up- or down-regulated genes with the nearest enhancers (H3K27ac peaks). Specifically, we first selected the up-regulated and down-regulated DEGs, and a set of randomly sampled stable genes with similar size (500 genes). Then for each gene set, we found the nearest enhancer (non-promoter H3K27ac peaks) and calculated the genomic distance. Then the cumulative distribution of the distances for the three sets of genes was plotted.

## ChIP-seq peak and dysregulated gene expression analysis

*ChIP-seq data processing and normalization.* CTCF ChIP-seq samples were taken across wild-type (WT) and *Matr3* KO conditions at days 0 and 4 of a target cell line. Sequencing reads from each sample were trimmed with Trimmomatic[101], aligned with Bowtie2[83], and subject to MACS2[86] peak calling (q < 0.10). The exact settings of each program are as follows: Trimmomatic (ILLUMINACLIP:-Truseq3.SE.fa:2:15:4 LEADING:20 TRAILING:20 SLIDINGWINDOW:4:15 MIN-LEN:25), Bowtie2 (-p 2 --phred33 -x mm10), MACS2 (callpeak -t X.bam -g mm -f BAM -q 0.10).

For a condition group (KOd0, KOd4, WTd0, WTd4) CTCF ChIP-seq samples were next normalized according to the following procedure. The intuition behind the whole normalization procedure is to normalize the samples such that all samples have the same background intensity level. (1) Let U be the union of all peaks from all samples of a condition group. (2) Let $U_f$ be the flank-regions of peaks in U, defined as the (-10,000 bp, 0 bp) upstream and (0 bp, +10,000 bp) downstream of each peak in U, not overlapping with any other peaks in U. $U_f$ gauges at the background region signal. (3) Using featureCount[102], we compute the number of reads in each genomic region of $U_f$ across samples. At the end, we produce matrix M, number of regions in $U_f$ x number of samples. (4) We feed M as input for DEseq2's estimateSizeFactor() calculation[92]. This function generates the size factors $q_1$, $q_2$, $q_3$, $q_4$ needed to normalize the original tracks. (5) Using samtools[103] sample function, original samples of Chip-seq were subsampled with each sample's calculated size factor $q_x$. With the subsampled reads, peak calling was repeated to define the final set of peaks for further analysis.

*Differential binding analysis and overlap analysis.* Each condition of a condition group (KOd0, KOd4, WTd0, WTd4) has at least 2 replicates. For differential binding analysis, we first define a set of common genomic regions between all 8 samples (2 replicate per condition × 4 conditions). The common regions are composed of the union of all 8 samples' MACS2-called peaks (from each sample's subsampled reads). A feature count table was created (genomic regions×8 samples). DESeq2[92] analysis was applied with the following factor design "factor(paste0(-phenotype$time, phenotype$state))", and skipping the estimateSizeFactors() step since samples have been already normalized by reads-downsampling. As a result, we add the line "sizeFactors(dds)<-rep(1,num_sample)" to the script. Then standard DEseq2 steps were followed as usual. At the end, differential regions are returned by DEseq2 for the following contrast groups: WTd0 vs. KOd0, and WTd0 vs. WTd4. The set of differential peaks from (1) WTd0_KOd0 group and (2) WTd0_WTd4 group are next compared to each other, and overlapped to produce a Venn diagram.

*ChIP-seq peak—RNA-seq differential gene association analysis.* Differential ChIP-seq peaks were associated with differentially expressed genes according to the following procedure. DESeq2 returned the differentially bound CTCF peak regions along with base mean, log2 fold-change and P-values. To make sure that the unit of comparison is consistently in gene space across ChIP-seq and RNA-seq datasets, we first mapped the differentially bound CTCF peaks to nearby genes by the following formula. For a contrast group, WTd0 vs. KOd0, a gene score is derived as follows. Each gene's CTCF differential binding score $s_g$, is computed as $s_g =$

$$\sum_{c \in g} f_c / n_c$$ where $c$ is each peak that maps to gene $g$ by ± 25 kb TSS criterion; $f_c$ is the $-\log_{10}$Pvalue of peak $c$; $n_c$ is the total number of genes that peak $c$ maps to (±25 kb TSS). A rank list of genes is produced from highest to lowest $s_g$. Next the rank-list is sorted into ventiles (20 bins). We overlapped all genes within each ventile with the differentially expressed genes from RNA-seq (WTd0 vs. KOd0). Finally, we plotted the overlap number per bin as seen in Fig. 7e, f. In the null hypothesis case, we do not expect to see an association between ChIP-seq peak-mapped genes and differentially expressed genes, and the overlap frequency should be uniform across all ventiles. Any deviation from the uniform distribution, with an over-representation towards the left of the plot, will indicate a non-random significant association. The minimum hypergeometric distribution score[104], available in the GOrilla package, was used to assess the significance of the association and reports a P-value.

## Chromatin CAPTURE experiment

*sgRNA design and cloning.* sgRNAs were designed to target the proximity of the Rad21 and CTCF binding site near *Mbd1* gene as well as to 10 kb upstream intergenic region that does not contain Rad21 or CTCF binding sites to serve as negative controls using the public tool (http://crispr.mit.edu/) for the mm10 mouse reference genome. A total of two sgRNAs per genomic region were cloned into the U6 promoter-driven lentiviral vector pSLQ1651-sgRNA(F + E)-sgGal4 (Addgene, #100549) by PCR amplification using a common reverse primer and unique forward primers containing the protospacer sequence[105]. The PCR amplicon and the sgRNA vector containing a mCherry reporter were digested by BstXI and XhoI. The digested DNA fragments were then purified, ligated to the digested sgRNA vector, and validated by Sanger sequencing. The protospacer sequence for each region are listed as follows with PAM sequence (NGG): *Mbd1*_prox_sgRNA 1, 5′–GGTTCCTTGCTTGGGAACCA (AGG)–3′; *Mbd1*_ prox _sgRNA 2, 5′–AACAAAGGCGGAATGTCTCC (TGG)–3′; *Mbd1*_upstream_control sgRNA 1, 5′–TAAGGGTGAAGCACACTAAA (TGG)–3′; *Mbd1*_upstream_control sgRNA 2, 5′– TGAATGGGCAAGCTCTCACT (TGG) – 3'.

*Cloning of CAPTURE vector.* To generate the pEF1a-dCas9-CBio-IRES-zsGreen1-puro vector, the dCas9-CBio-IRES-zsGreen1 fragment was amplified from pLVX-EF1a-dCas9-CBio-IRES-zsGreen1 (Addgene #138418) as the template and cloned into XbaI and SmaI double-digested pEF1a-FB-dCas9-puro vector (Addgene #100547) by In-Fusion HD Cloning Kit (Clontech).

*Derivation and maintenance of CAPTURE cell lines.* WT and KO MEL cell lines were electroporated with ClaI-linearized pEF1a-dCas9-CBio-IRES-zsGreen1-puro vector using a BTX ECM830 square electroporator (BTX Harvard Apparatus) with 1 pulse at 250 V for 15msec. After 48 hours, puromycin at 5 µg/mL was added to culture media and cells were subsequently maintained with puromycin to select for stably-expressing cell lines. At 7 days post-electroporation, the top 10% zsGreen1-positive MEL cells were sorted using FACSAria cell sorters (BD Biosciences).

*Lentivirus production and transduction.* Lentiviruses containing sgRNAs were packaged into HEK293T cells. 6.5 µg of psPAX2, 3.5 µg of VSV-g, and 10 µg of sgRNA lentiviral vectors were co-transfected into HEK293T cells in 10-cm dishes with 80 µg of branched polyethylenimine (PEI). Lentiviruses were collected by harvesting the supernatant 48-72 h post-transfection. dCas9-CBio-zsGreen1 expressing and puromycin-resistant MEL cells were transduced with sgRNA-expressing lentiviruses in 6-well plates. To maximize sgRNA expression, the top 5% mCherry-positive and zsGreen1-positive cells were sorted 48 h post-transduction.

*CAPTURE ChIP-qPCR assay.* In total $5 \times 10^6$ WT and KO MEL cells transduced with *Mbd1* proximal region-targeting and negative control sgRNAs were used. Cells were cross-linked with 1% formaldehyde for 10 min and quenched with 0.125 M of glycine for 5 min. After washing with PBS, cells were lysed in 1 mL cell lysis buffer (25 mM Tris-HCl pH 7.4, 85 mM KCl, 0.25% Triton X-100, freshly added 1 mM DTT and 1:200 protease inhibitor cocktail (Sigma)) and rotated for 30 min at 4 °C. Nuclei were collected by centrifugation at 2,500 x g for 5 min at 4 °C. Nuclear pellets were resuspended in 0.5% SDS nuclear lysis buffer (50 mM Tris-HCl, pH 8.1, 10 mM EDTA, 0.5% SDS, freshly added 1 mM DTT and 1:200 protease inhibitor cocktail) and chromatin was sonicated to an average size 200 to 500 bp on the Branson Sonifier 450 ultrasonic processor (20% amplitude, 0.5 sec ON, 1 sec OFF, for 20 sec). Supernatant containing soluble chromatin was transferred to a new tube. Final concentration 300 mM NaCl and 1% Triton X-100 was added to 400 µL of supernatant, followed by rotation with 20 µL of MyOne Streptavidin T1 Dynabeads (Thermo Scientific) at 4 °C. After overnight incubation,

Dynabeads were washed twice with 500 µL of 2% SDS, twice with 500 µL of RIPA buffer with 0.5 M NaCl (10 mM Tris-HCl, 1 mM EDTA, 0.1% sodium deoxycholate, 0.1% SDS, 1% Triton X-100, pH 8.0), twice with 500 µL of LiCl buffer (250 mM LiCl, 0.5% NP-40, 0.5% sodium deoxycholate, 1 mM EDTA, 10 mM Tris-HCl pH 8.0), and twice with 500 µL of TE buffer (10 mM Tris-HCl, 1 mM EDTA, pH 8.0). Chromatin was eluted in SDS elution buffer (1% SDS, 10 mM EDTA, 50 mM Tris-HCl, pH 8.0) and reverse cross-linked at 65 °C with shaking at 1050 rpm overnight. ChIP DNA was incubated with RNase A 5 µg/mL (Thermo Scientific) and 0.2 mg/mL proteinase K (Ambion) at 37 °C for 30 min and 2 hours, respectively, and purified using QIAquick Spin columns (Qiagen). Quantitative RT-PCR (qPCR) was performed using the iQ SYBR Green Supermix (Bio-Rad). Primers are listed as follows: Mbd1_prox_forward: 5′–CGGGTACCAATCCTGAAGAA–3′; Mbd1_prox_reverse: 5′–AGTCGCTCCGGACACAAG–3′; Mbd1_upstream_control_forward: 5′–CAGGTGGCCAGCTTAATAAAA–3′; Mbd1_upstream_control_reverse: 5′–CAAGTGAGAGCTTGCCCATT–3′; Negative_control_forward: 5′–CCTCTGATTGATCCCCAGCA–3′; Negative_control_reverse: 5′–ACACCGACTGACTGCATGAG–3′.

*CAPTURE Western Blot assay.* In total 1 to $2 \times 10^8$ WT and KO MEL cells transduced with Mbd1 proximal region-targeting and negative control sgRNAs were used. Cells were cross-linked with 2% formaldehyde for 10 min and quenched with 0.125 M of glycine for 5 min. After washing with PBS, cells were lysed in 1 mL cell lysis buffer (25 mM Tris-HCl pH 7.4, 85 mM KCl, 0.25% Triton X-100, freshly added 1 mM DTT and 1:200 protease inhibitor cocktail (Sigma)) and rotated for 30 min at 4 °C. Cell nuclei were collected by centrifugation at $2,500 \times g$ for 5 min at 4 °C. Nuclei were resuspended in 500 µL cell lysis buffer with 1 µL 0.5 µg/mL RNase A and rotated at 37 °C for 30 min to degrade chromatin associated-RNAs. Nuclei were centrifuged at $2,500 \times g$ for 5 min at 4 °C. Nuclear pellets were resuspended in 400 µL 4% SDS nuclear lysis buffer (50 mM Tris-HCl pH 7.4, 10 mM EDTA, 4% SDS, freshly added 1 mM DTT and 1:200 protease inhibitor cocktail) and incubated at room temperature for 10 min. Nuclei suspension was mixed with 1.2 mL freshly prepared 8 M urea buffer (10 mM Tris-HCl pH 7.4, 1 mM EDTA, 8 M Urea) and centrifuged at $16,100 \times g$ for 25 min at 22 °C. The samples were washed twice more in 0.4 mL nuclear lysis buffer and mixed with 1.2 mL 8 M urea buffer, followed by centrifugation at $16,100 \times g$ for 25 min at 22 °C. Pelleted chromatin was then washed twice with nuclear lysis buffer followed by two washes with modified cell lysis buffer (25 mM Tris-HCl pH 7.4, 10 mM KCl, 0.25% Triton X-100) to remove residual urea and SDS, respectively. Chromatin pellet was resuspended in 800 µL IP binding buffer without NaCl (20 mM Tris-HCl pH 7.5, 1 mM EDTA, 0.1% NP-40, 10% glycerol, freshly added proteinase inhibitor). Chromatin suspension was then subjected to sonication to an average size of ~500 bp on the Branson Sonifier 450 ultrasonic processor (10% amplitude, 0.5 s ON, 1 s OFF, for 20 s). Fragmented chromatin was centrifuged at $16,100 \times g$ for 10 min at 4 °C. Supernatant was combined and final concentration of 150 mM NaCl was added to the sheared chromatin. To prepare the streptavidin beads for affinity purification, 50 µL of streptavidin agarose slurry (Life Technologies) was washed 3 times in 1 mL of IP binding buffer and added to soluble chromatin. After overnight rotation at 4 °C, streptavidin beads were collected by centrifugation at 800 x g for 3 min at 4 °C. The beads were wash twice with 1 mL of 2% SDS, twice with 1 mL of RIPA buffer with 0.5 M NaCl, twice with 1 mL of LiCl buffer, and twice with 1 mL of TE buffer. The chromatin was resuspended in 15 µL of RIPA buffer (50 mM Tris-HCl, 1% NP-40, 0.25% sodium deoxycholate, 150 mM NaCl, 0.1% SDS, 2 mM EDTA) and incubated with 1 µL of Benzonase nuclease (Sigma) overnight at 4 °C. The following morning, 5 µL 4× XT sample loading buffer containing 1.25% 2-mercaptoethanol was added to the sample followed by incubation at 95 C for 20 min followed by 5 min incubation on ice. Protein sample was centrifuged at $12,000 \times g$ for 10 min at 4 °C. The protein sample was loaded onto NuPAGE™ 4–12% Bis-Tris gels (Invitrogen) and run with 1× MOPS running buffer (Invitrogen) and transferred to Amersham Hybond P 0.45 PVDF blots (GE Healthcare #10600023). The blots were incubated with primary antibodies against Matrin-3 (Santa Cruz Biotechnology, sc-81318) and Histone H3 (Abcam, ab1791) with 1:100 and 1:3000 dilutions, respectively, in 5% non-fat milk in TBS/T (20 mM Tris-HCl, pH7.5, 150 mM NaCl, 0.1% Tween-20) at 4 °C overnight. After washing 3 times with TBS/T, the blots were incubated with secondary antibodies (Cell Signaling Technologies, anti-Mouse-HRP CST 7076, anti-Rabbit-HRP CST 7074) with 5% non-fat milk in TBS/T at 1:3000 dilutions for 1 h at room temperature. The blots were then washed 3 times with TBS/T and developed using Plus-ECL (PerkinElmer). Densitometry quantification was performed using ImageJ software.

**Reporting summary**. Further information on research design is available in the Nature Research Reporting Summary linked to this article.

## Data availability
The data that support this study are available from the corresponding author upon reasonable request. Hi-C, ChIP-seq, ATAC-seq, and RNA-seq data sets generated this study have been deposited in the GEO database, under accession code GSE181234. Proteomic data are available via ProteomeXchange with identifier PXD028867. Source data are provided with this paper.

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

## Acknowledgements

We thank Richard Young and Isaac Klein for advice and assistance with nuclear body screening, Xin Liu for designing the CAPTURE assay, Nan Liu for helping with CUT&RUN procedure, and Deniz Ozata for planning Hi-C experiment. We appreciate David Pellman for sharing the spinning disk confocal microscope. We also would like to thank Meeta Mistry of the Harvard Chan Bioinformatics Core, Harvard T.H. Chan School of Public Health, Boston, MA for assistance with ChIP-seq analysis. This work was supported by the Howard Hughes Medical Institute (HHMI to S.H.O. and J.D.); National Heart, Lung, and Blood Institute (HL119099 and HL032262 to S.H.O.); National Human Genome Research Institute (HG009663 to G.-C.Y.); National Cancer Institute and National Institute of Diabetes and Digestive and Kidney Diseases (R01CA230631 and R01DK111430 to J.X.); a fellow award from the Leukemia & Lymphoma Society to H.J.C.

## Author contributions

H.J.C. and S.H.O. conceptualized and designed research. H.J.C., O.U., T.L., Z.T. and G.B. performed the experiments. H.J.C., O.U., Y.K., Q.Z., Z.T. and G.B. analyzed the data. H.J.C., O.U., Y.K., T.L., Q.Z., Z.T., G.B., J.X., G.-C.Y., J.D. and S.H.O. interpreted the data. H.J.C. and S.H.O wrote the manuscript with input from all authors.

## Competing interests

The authors have no competing interests.
