## [Peer Review File · Nature Communications]

Inner nuclear protein Matrin-3 coordinates cell differentiation by stabilizing chromatin architectureEditorial Note: This manuscript has been previously reviewed at another journal that is not operating a transparent peer review scheme. The manuscript was considered suitable for publication without further review at Nature Communications.

REVIEWER COMMENTS

Reviewer #1 (Remarks to the Author):

Cha et al present a manuscript in which they discuss the effect of loss of the nuclear matrix protein Matr3. The authors show that Matr3 loss leads to precocious differentiation in MEL cells. This leads to accelerated differentiation, as shown by the increased expression of the hemoglobin subunit beta-globin. Matr3 loss is also associated with less well-defined heterochromatin spots as measured by immunofluorescence analysis. This ultrastructural change is accompanied by a mild effect on 3D genome organization as measured by Hi-C. There is an increase in interactions between B compartments and a decrease in interactions between A-compartment.

The authors perform a BioID experiment in which they show that Matr3 interacts with CTCF and ESCO2. They also perform SMC1A pull-down and show an interaction with Matr3. They perform ChIPseq of RAD21 and CTCF and show that there is decreased binding of these factors in the KO. The authors perform some functional assays that suggest that the Mbd1 gene is regulated by Matr3 through CTCF/cohesin. In the final figure the authors integrate the Hi-C data with ChIPseq and RNAseq data to link Matr3 to the 3D genome and gene expression.

This final analysis is the weakest part of the manuscript. The suggestion is (also indicated by the model figure) is that Matr3 is important for CTCF/cohesin looping. However, not a single example in which looping is affected is shown; only highly abstract analyses showing only p-values are presented to claim that Matr3 has an effect. The authors should also show effect sizes for all the analyses. The authors claim that weakly bound CTCF sites are most strongly affected, which would make sense, as weak sites are more easily lost. Because these are weak sites, these sites also show weaker insulation which makes sense. Why is the difference between WT and KO not shown, but only KO? In general the effects observed on the 3D genome are very weak. The author should show the results of the Hi-C analyses for all the replicate experiments of the Hi-C so that readers can gauge the inter-experiment variability. The authors should also show more examples of ChIPseq tracks also in combination with the Hi-C data to back up their claims.

For the BioID experiment: the method section mentions that the control is BirA. Is this nuclear BirA? If not the author should repeat the experiment with a BirA with an NLS. The authors should show volcano plots of Matr3-BirA over nuclear BirA. The number of peptides seems quite low. Related to this, why was the cohesin complex not identified?

Other points:

* in the saddle plots are the score for a single bin or all AA and BB over AB interactions, because the score seems to be quite high for the combination of all A compartments. I fail to see how a difference of -0.4 to 0.4 (-0.8, i.e. less than 2-fold) can lead to a score of 5.34.

* Figure 3I show $-\log p$ -values of 600. Without effect sizes these p-values are largely meaningless. Also, is this natural log or \log_{10} ; this seems to be confusing throughout the manuscript.

* Loss of cohesin Matr3 interaction in the Stag2 KO is hardly a validation there is still Stag1 cohesin, so this may be an interesting observation, that needs further validation, but cannot be used as validation of the cohesin interaction.

* Fig. 5B according to the genome browser there is a Mbd1 isoform that overlaps with the CTCF/cohesin binding site. This could mean that the CRISPR KO may inadvertently also knock-out the

promoter. Can the authors confirm that this is not the case?

* The authors seem to be convinced that CTCF mediates promoter-enhancer interactions, however, the current literature seems to be more inclined towards a role for CTCF in enhancer blocking. Also the authors cite a paper (ref. 46) for the role of cohesin in promoter-enhancer looping, which explicitly shows examples of cohesin independent promoter-enhancer looping.

Reviewer #2 (Remarks to the Author):

In this manuscript, Cha and colleagues elucidate and decipher the role of Matrin3, a nuclear protein, on erythroid differentiation, gene expression, and genome organization. They demonstrate that loss of Matrin3 results in premature differentiation of erythroid cells which is associated with loss of CTCF and RAD21 occupancy at a subset of sites. They show that Matrin3 physically interacts with CTCF and components of the cohesin ring complex, and also deletion of CTCF binding sites upstream of an erythroid-relevant gene, *Mbd1*, recapitulates gene expression changes found upon loss of Matrin3.

Overall - the manuscript is well-written, data well-presented, and analysis rigorous. The work will be of interest to a broad range of scientists. Below are a few suggestions for the authors to consider:

The authors demonstrate that B compartment intra-TAD interactions are weakened upon Matrin3 depletion (and conversely A compartment intra-TAD interactions are strengthened) (Fig. 2I). A similar trend is observed upon Mel1 differentiation comparing the undifferentiated to the differentiated state (Fig. 3C). Given the authors' conclude that loss of Matrin3 results in precocious differentiation, what is the correlation between change in intra-tad interactions upon Mel1 differentiation versus Matrin3 loss. For example, are the TADs that are changing the most (or least) upon Mel1 differentiation also changing (the most or least) upon Matrin3 loss? Alternatively, this could be evaluated on a tad-by-tad basis in a scatter plot format.

The authors probe a specific CTCF/cohesin-occupied regulatory region upstream of *Mbd1* and show deletion of that region reduces *Mbd1* expression. It would be helpful if the authors were able to better describe this regulatory region - specifically is it at a (sub-)TAD boundary or loop anchor - which would be relevant to the model proposed. As the authors know, CTCF is known to bind at locations outside of TAD boundaries/loop anchors. Relatedly, does knockdown of CTCF and/or RAD21 result in a down-regulation of *Mbd1*? Also, it is possible that CTCF/cohesin occupancy drives the expression of *Mbd1* from this element, but other factors which bind at that element could also be important. The authors may want to include this limitation/caveat in their interpretation of the result.

What is the expression pattern of Matrin3 during normal erythroid development? And similarly, is there any change in the strength of the Interaction between CTCF and Matrin3 in differentiated Mel1 cells as opposed to undifferentiated ones?

The authors show changes in CTCF, Rad21 occupancy as well HP1alpha staining. It would be important to ensure that deletion of Matrin3 does not affect the expression of these proteins (and other subunits in the cohesin complex). This information should be readily available from the gene expression datasets generated.

A subset of CTCF + Cohesin sites are co-occupied across the genome (often at TAD boundaries). Does Matrin3 loss affect co-occupied regions versus CTCF-only or Rad21-only regions? (Or vice-versa?)

Minor points:

The authors show that GATA motifs are enriched in the "newly" gained ATAC-seq peaks upon Matrin3,

Were any regions less accessible upon Matrin3? And if so, did it relate to the biology elucidated?

Statistical analysis on 1E (specifically between parental and KO at each day) would be helpful.

A western blot showing the expression of exogenous Matrin3 in the experiment relevant to Fig 1D would be helpful.

3F - typo on the figure - should be "insulation".

The authors may consider additional annotation of the figures to improve readability and a legend for the model.

The authors may consider violin plots to replace some of their box plots - as it might show the differences in the distribution of the data a bit better (some examples include 1C, 2I, 3C)

Point-by-point response/response to referees

We thank the referees for their comments and constructive suggestions. Below (in blue) are detailed point-by-point responses to the reviewer comments.

Reviewer #1 (Remarks to the Author):

Cha et al present a manuscript in which they discuss the effect of loss of the nuclear matrix protein Matr3. The authors show that Matr3 loss leads to precocious differentiation in MEL cells. This leads to accelerated differentiation, as shown by the increased expression of the hemoglobin subunit beta-globin. Matr3 loss is also associated with less well-defined heterochromatin spots as measured by immunofluorescence analysis. This ultrastructural change is accompanied by a mild effect on 3D genome organization as measured by Hi-C. There is an increase in interactions between B compartments and a decrease in interactions between A-compartment.

The authors perform a BioID experiment in which they show that Matr3 interacts with CTCF and ESCO2. They also perform SMC1A pull-down and show an interaction with Matr3. They perform ChIPseq of RAD21 and CTCF and show that there is decreased binding of these factors in the KO. The authors perform some functional assays that suggest that the Mbd1 gene is regulated by Matr3 through CTCF/cohesin. In the final figure the authors integrate the Hi-C data with ChIPseq and RNAseq data to link Matr3 to the 3D genome and gene expression.

This final analysis is the weakest part of the manuscript. The suggestion is (also indicated by the model figure) is that Matr3 is important for CTCF/cohesin looping. However, not a single example in which looping is affected is shown; only highly abstract analyses showing only p-values are presented to claim that Matr3 has an effect. The authors should also show effect sizes for all the analyses. The authors claim that weakly bound CTCF sites are most strongly affected, which would make sense, as weak sites are more easily lost. Because these are weak sites, these sites also show weaker insulation which makes sense. Why is the difference between WT and KO not shown, but only KO?

We appreciate the reviewer for taking the time to review our manuscript. One misunderstanding by the reviewer is that we performed immunoprecipitation of biotinylated Matr3 rather than a BioID experiment. We will explain in more detail below.

As the reviewer pointed out, our data on chromatin occupancy of CTCF/cohesin strongly suggested that Matr3 loss might affect chromatin looping. To address the reviewer's concern, we called the differential loops in Matr3 KO compared to parental cells and differential loops during MEL cell differentiation. Among them, 3573 and 3575 differential loops were lost in Matr3 KO and during differentiation, respectively. Consistent with the findings associated with CTCF/cohesin, 2996 (84%) of the loops overlapped. We add this new result to Figure 6I. The random overlap for the lost differential loops in the permutation test was 212 out of 3650 (6%, $p < 1e-5$). Also, we now showed effect sizes for all analyses.

Specific loops obtained from differential loop analysis

We thank the reviewer for understanding and agreeing to our work on weakly bound CTCF sites. We regret that our original manuscript was not clear. In fact, the reviewer is mistaken about what is shown. CTCF regions of WT were split into a set of maintained (common) and a set of lost (uniq) regions when *Matr3* was knocked out, and insulation of those regions was compared. In the revised manuscript, we explain this more clearly in the legend of Figure 6.

In general the effects observed on the 3D genome are very weak. The author should show the results of the Hi-C analyses for all the replicate experiments of the Hi-C so that readers can gauge the inter-experiment variability. The authors should also show more examples of ChIPseq tracks also in combination with the Hi-C data to back up their claims.

We now added the Hi-C analysis results of all replicates to Figures S1-S2. Also, we show examples of ChIP-seq tracks combined with Hi-C data in Figure S5G.

Hi-C analysis replicates (A-E : rep.1, A'-E' : rep. 2)

Examples of regions showing altered chromatin interaction loops with reduced CTCF and Rad21 binding in Matr3 KO compared to control.

For the BioID experiment: the method section mentions that the control is BirA. Is this nuclear BirA? If not the author should repeat the experiment with a BirA with an NLS. The authors should show volcano plots of Matr3-BirA over nuclear BirA. The number of peptides seems quite low. Related to this, why was the cohesin complex not identified?

We performed immunoprecipitation of biotinylated Matr3, not a BioID experiment. BioID is a proximity labeling method that uses mutant biotin ligase to biotinylate proximal proteins and thus can detect weak and/or transient interactions. Here, we used *in vivo* biotinylation of Matr3 for affinity purification. This method is previously described: Kim et al., Nat. Protoc. 4, 506–517 (2009). Briefly, the *in vivo* biotinylation experiment is based on a short biotinylated peptide fused to Matr3, which serves as an *in vivo* substrate for BirA. Prior to expressing Matr3 fused with this biotinylated peptide, a cell line stably expressing BirA was established and used as a control for background signal during affinity purification.

For data analysis, we were advised by Steve Gygi of the Taplin Mass Spectrometry Facility at Harvard Medical School that it is best to only consider proteins found in the targeted IP and not in the control; therefore, there is no comparison. We generated a volcano plot using the sum intensity. Data were log 2 transformed, normalized by the average and distribution width, and then used for statistics.

We speculate that the cohesin complex was not identified because of its weak binding to Matr3. Clones with sub-endogenous Matr3 expression levels were selected for affinity purification to minimize interference with the endogenous protein complex by the tagged protein as described (Kim et al., Nat. Protoc., 2009). Therefore, a protein that binds weakly/transiently to Matr3 may be missing. In Figure 4B we immunoprecipitated endogenous Matr3 and confirmed binding to cohesin.

Other points:

* in the saddle plots are the score for a single bin or all AA and BB over AB interactions, because the score seems to be quite high for the combination of all A compartments. I fail to see how a difference of -0.4 to 0.4 (-0.8, i.e. less than 2-fold) can lead to a score of 5.34.

We apologize for the lack of detail. For saddle plots for each experiment (Figures 2F and 3A), we calculated top 20% quantile of sorted EV1 values and divided AA/AB and BB/AB values. Scores on the plots represent the average value of all 20% highest EV1. In Figures 2G and 3B, the difference was calculated as log2 ratio of average interaction intensity (obs/exp) in Matr3 KO and control. We added this information to the legend of Figure 2.

* Figure 3I show -log p-values of 600. Without effect sizes these p-values are largely meaningless. Also, is this natural log or log10; this seems to be confusing throughout the manuscript.

A table containing the raw number and percentage of the sequences used for motif analysis is now added to Table S1 to reveal effect sizes.

Name	log p-value	# Target Sequences with Motif	% of Targets Sequences with Motif	# Background Sequences with Motif	% of Background Sequences with Motif
Gata4 (Zf)	-5.46E+02	1034	27.54%	4115.3	8.94%
Gata6 (Zf)	-5.33E+02	967	25.76%	3701.1	8.04%
Gata3 (Zf)	-5.29E+02	1276	33.99%	6097.6	13.25%
Gata2 (Zf)	-5.21E+02	823	21.92%	2786.2	6.05%
Gata1 (Zf)	-5.09E+02	773	20.59%	2518.2	5.47%
PU.1 (ETS)	-1.94E+02	489	13.03%	2227.6	4.84%
ETS1 (ETS)	-1.75E+02	769	20.48%	4712.9	10.24%

We apologize for the confusion. This is a natural log. We added this to the Figure 3 legend and clarified the log base throughout the manuscript or figures.

* Loss of cohesin Matr3 interaction in the Stag2 KO is hardly a validation there is still Stag1 cohesin, so this may be an interesting observation, that needs further validation, but cannot be used as validation of the cohesin interaction.

We edited the manuscript based on this advice.

* Fig. 5B according to the genome browser there is a Mbd1 isoform that overlaps with the CTCF/cohesin binding site. This could mean that the CRISPR KO may inadvertently also knock-out the promoter. Can the authors confirm that this is not the case?

We aimed to specifically delete the cis element to which CTCF/cohesin binds. Because this binding site is close to the TSS, it is difficult to rule out the possibility that other factors that bind at that element could also be important. Therefore, as suggested by reviewer #2, we added this limitation to the interpretation of the result.

* The authors seem to be convinced that CTCF mediates promoter-enhancer interactions, however, the current literature seems to be more inclined towards a role for CTCF in enhancer blocking. Also the authors cite a paper (ref. 46) for the role of cohesin in promoter-enhancer looping, which explicitly shows examples of cohesin independent promoter-enhancer looping.

We agree that CTCF has multiple roles in regulating chromosomal interactions and we think that emerging studies on this will help build an integrated model that can better explain its function. In this paper, we focused on weakly bound CTCF sites affected by Matr3 loss and investigated changes in expression of deregulated genes, which include all genes affected by altered enhancer function (both blocking and mediating).

We cited ref. 46 as an example of changes in chromatin occupancy of cohesin that occur during differentiation (lineage-specific loops established by cohesin in epidermal progenitor cells were not present in the pluripotent state). We edited the text to reduce confusion.

Reviewer #2 (Remarks to the Author):

In this manuscript, Cha and colleagues elucidate and decipher the role of Matr3, a nuclear protein, on erythroid differentiation, gene expression, and genome organization. They demonstrate that loss of Matr3 results in premature differentiation of erythroid cells which is associated with loss of CTCF and RAD21 occupancy at a subset of sites. They show that Matr3 physically interacts with CTCF and components of the cohesin ring complex, and also deletion of CTCF binding sites upstream of an erythroid-relevant gene, Mbd1, recapitulates gene expression changes found upon loss of Matr3.

Overall - the manuscript is well-written, data well-presented, and analysis rigorous. The work will be of interest to a broad range of scientists. Below are a few suggestions for the authors to consider:

The authors demonstrate that B compartment intra-TAD interactions are weakened upon Matr3 depletion (and conversely A compartment intra-TAD interactions are strengthened) (Fig. 2I). A similar trend is observed upon Mel1 differentiation comparing the undifferentiated to the differentiated state (Fig. 3C). Given the authors' conclude that loss of Matr3 results in precocious differentiation, what is the correlation between change in intra-tad interactions upon Mel1 differentiation versus Matr3 loss. For example, are the TADs that are changing the most (or least) upon Mel1 differentiation also changing (the most or least) upon Matr3 loss?

Alternatively, this could be evaluated on a tad-by-tad basis in a scatter plot format.

We thank the reviewer for the time taken to review our manuscript and appreciation of our work. As suggested, we compared changes in intra-TAD interaction frequency following MEL cell differentiation and *Matr3* loss. Consistent with other findings, TADs with significantly altered interactions in *Matr3* KO and during differentiation overlapped significantly. We added this result to Figure S2C.

TADs with significantly altered interactions in *Matr3* KO and during differentiation were compared. TADs with increased interaction in compartment A significantly overlapped (2.03 fold over enriched compared to expected, $p < 1e-100$), and TADs with decreased interaction in compartment B also overlapped significantly (1.98 fold over enriched compared to expectations, $p < 1e-100$).

The authors probe a specific CTCF/cohesin-occupied regulatory region upstream of *Mbd1* and show deletion of that region reduces *Mbd1* expression. It would be helpful if the authors were able to better describe this regulatory region - specifically is it at a (sub-)TAD boundary or loop anchor - which would be relevant to the model proposed. As the authors know, CTCF is known to bind at locations outside of TAD boundaries/loop anchors. Relatedly, does knockdown of CTCF and/or RAD21 result in a down-regulation of *Mbd1*? Also, it is possible that CTCF/cohesin occupancy drives the expression of *Mbd1* from this element, but other factors which bind at that element could also be important. The authors may want to include this limitation/caveat in their interpretation of the result.

As suggested by the reviewer, we now add Hi-C contact matrices around the *Mbd1* gene in Figure S4A (Dashed box indicates the CTCF/cohesin-occupied regulatory region upstream of *Mbd1*). *Mbd1* was indeed located near a loop anchor and the insulation became weaker in *Matr3* KO cells.

Hi-C contact matrices at 5 kb resolution and compartments near the Mbd1 gene. Dashed box indicates the CTCF/Rad21-occupied region upstream of Mbd1.

We knocked down CTCF and Rad21 using siRNA and measured Mbd1 expression compared to a scrambled control. Mbd1 expression was not significantly altered. Because there is residual expression of CTCF and Rad21, the possibility that CTCF and Rad21 still bind upstream of Mbd1 cannot be excluded. Therefore, we find it difficult to draw a definitive conclusion from a knockdown experiment. Also, we agree with the reviewer's comment that other factors that bind to this element could also be important, and included this limitation in the manuscript.

(Left) Reduced CTCF and Rad21 expression in MEL cells was assessed by Western blot. β -actin was used as a Western blot control. (Right) Mbd1 mRNA was not significantly altered in CTCF/Rad21 siRNA treated cells compared to a scramble control.

What is the expression pattern of Matr3 during normal erythroid development? And similarly, is there any change in the strength of the Interaction between CTCF and Matr3 in differentiated Mel1 cells as opposed to undifferentiated ones?

Matr3 is continuously expressed during normal erythroid development. We measured RNA and protein level expression in MEL cells and Matr3 expression was similar in untreated and differentiated cells (day 4). Matr3 was also consistently expressed during

ex vivo erythropoiesis of human cd34+ cells. Similarly, the strength of the interaction between CTCF and Matr3 was comparable during MEL cell differentiation.

(Top) Matr3 expression levels were compared in untreated and differentiated MEL cells using RNA-seq data and Western blot. (Bottom, left) Matr3 expression during human CD34+ differentiation was assessed by Western blot. (Bottom, right) Endogenous Matr3 protein was immunoprecipitated and its interaction with CTCF/Rad21 was examined in untreated and differentiated MEL cells by Western blot.

The authors show changes in CTCF, Rad21 occupancy as well HP1alpha staining. It would be important to ensure that deletion of Matr3 does not affect the expression of these proteins (and other subunits in the cohesin complex). This information should be readily available from the gene expression datasets generated.

Matr3 deletion does not affect the expression of HP1 α and subunits of the cohesin complex. We added these results to the text and Figures S1F and S3C.

Expression levels of HP1α and subunits of the cohesin complex were compared in parental and Matr3 KO MEL cells using RNA-seq data.

A subset of CTCF + Cohesin sites are co-occupied across the genome (often at TAD boundaries). Does Matr3 loss affect co-occupied regions versus CTCF-only or Rad21-only regions? (Or vice-versa?)

We compared 5539 CTCF regions and 3429 Rad21 regions whose occupancy decreased upon Matr3 deletion. 422 regions overlapped. Thus, Matr3 loss affects both co-occupied regions and CTCF/Rad21-only regions. The larger number of the latter regions may be because the co-occupied regions tend to be strong binding regions, whereas Matr3 has more impact on the weak binding regions.

Minor points:

The authors show that GATA motifs are enriched in the “newly” gained ATAC-seq peaks upon Matr3. Were any regions less accessible upon Matr3? And if so, did it relate to the biology elucidated?

In Matr3 KO cells compared to parental cells, 339 regions were less accessible while 4918 regions were newly accessible. Motifs such as Gata4(Zf)/1e-237, Gata6(Zf)/1e-231, and Gata3(Zf)/1e-229 were enriched in the gained ATAC regions, whereas EHF(ETS)/1e-4, SpiB(ETS)/1e-3, and ELF3(ETS)/1e-3 motifs were enriched in the less accessible regions (numbers shown are p-values). This result is consistent with the fact that those ETS motifs are more accessible in undifferentiated MEL cells.

Statistical analysis on 1E (specifically between parental and KO at each day) would be helpful.

As suggested by the reviewer, we added the p-values from the t-test to the Figure 1E legend. P-values obtained from the t-test between parental and Matr3 KO cells at each day were 9.69E-02, 6.83E-03, and 3.65E-07, respectively.

A western blot showing the expression of exogenous Matr3 in the experiment relevant to Fig 1D would be helpful.

Western blot showing expression was added to FigureS1C. Here we also add the original membrane before cropping the image (lanes #1 and #13 were cropped for the figure).

Matr3 expression in parental cells and Matr3 KO cells rescued with full-length Matr3 cDNA is shown by Western blot. β -actin was used as a control.

3F - typo on the figure - should be "insulation".

We fixed this typo. Thank you.

The authors may consider additional annotation of the figures to improve readability and a legend for the model.

As suggested by the reviewer, we added annotations and descriptions to the figures and a model legend. We also moved figures that provide similar messages to the supplement for better readability.

The authors may consider violin plots to replace some of their box plots - as it might show the differences in the distribution of the data a bit better (some examples include 1C, 2I, 3C)

We replaced the box plots with violin plots as suggested by the reviewer.

REVIEWERS' COMMENTS

Reviewer #1 (Remarks to the Author):

The authors have addressed most of my comments. They seem to have ignored my point regarding Figure 6G. They state:

"Chromatin insulation was reduced at the boundaries containing weak CTCF and Rad21 sites that were lost in the absence of Matr3, and more interactions were observed across the domain boundaries (Figures 6G-H and S5I-J)."

This statement is suggestive, as it seems to suggest that insulation is reduced upon KO of Matr3, however, this is not what is shown. What is shown is that insulation is, compared to the strong CTCF sites, is lower. This is a trivial observation: strong CTCF sites show stronger insulation, weak sites weaker insulation.

The authors should include a plot in which they show the differential insulation signal of the Matr3 KO Hi-C data over the parental Hi-C data to show the effect of CTCF loss on insulation following Matr3 KO.

Reviewer #2 (Remarks to the Author):

The authors have submitted a revised manuscript and addressed my comments, and I look forward to seeing the paper in its published form in the near future. A minor point -(new) Supp Figure 2C is only called out in the main figure legends. I may suggest it is called out in the main text, and include in the text and/or methods what "the expected" is in the comparison.

Point-by-point response

Reviewer #1 (Remarks to the Author):

The authors have addressed most of my comments. They seem to have ignored my point regarding Figure 6G. They state:

“Chromatin insulation was reduced at the boundaries containing weak CTCF and Rad21 sites that were lost in the absence of Matr3, and more interactions were observed across the domain boundaries (Figures 6G-H and S5I-J).”

This statement is suggestive, as it seems to suggest that insulation is reduced upon KO of Matr3, however, this is not what is shown. What is shown is that insulation is, compared to the strong CTCF sites, is lower. This is a trivial observation: strong CTCF sites show stronger insulation, weak sites weaker insulation.

The authors should include a plot in which they show the differential insulation signal of the Matr3 KO Hi-C data over the parental Hi-C data to show the effect of CTCF loss on insulation following Matr3 KO.

We thank the reviewer for the positive comment and detailed explanation of the question. To address the reviewer’s concern, we compared interaction pile-up maps of Hi-C data from Matr3 KO and parental cells at sites of CTCF or Rad21 loss. Indeed, chromatin insulation was reduced in Matr3 KO compared to parental cells, and this data is now added to Figure S6a-b'.

Interaction pile-up maps of Hi-C data from parental and Matr3 KO cells at the boundaries defined by altered CTCF and Rad21 sites.

Reviewer #2 (Remarks to the Author):

The authors have submitted a revised manuscript and addressed my comments, and I look forward to seeing the paper in its published form in the near future. A minor point -(new) Supp Figure 2C is only called out in the main figure legends. I may suggest it is called out in the main text, and include in the text and/or methods what "the expected" is in the comparison.

We are grateful to the reviewer for the positive comment. We now added the description of Figure S2C to the main text and included what “the expected” is in the Methods. “The expected” was the expected number of overlapping TADs at a random setting. It was calculated using the hyper-geometrical distribution.